# Small-molecule inhibitors identify the RAD52-ssDNA interaction as critical for recovery from replication stress and for survival of BRCA2 deficient cells

Sarah R Hengel[1], Eva Malacaria[2], Laura Folly da Silva Constantino[3], Fletcher E Bain[1], Andrea Diaz[1], Brandon G Koch[1], Liping Yu[1,4], Meng Wu[1,3,5], Pietro Pichierri[2], M Ashley Spies[1,3]*, Maria Spies[1]*

[1]Department of Biochemistry, University of Iowa, Iowa City, United States; [2]Department of Environment and Health, Section of Experimental and Computational Carcinogenesis, Istituto Superiore di Sanita, Rome, Italy; [3]Division of Medicinal and Natural Products Chemistry, Department of Pharmaceutical Sciences and Experimental Therapeutics, University of Iowa, Iowa City, United States; [4]NMR Core Facility, Carver College of Medicine, University of Iowa, Iowa City, United States; [5]High Throughput Screening Facility, University of Iowa, Iowa City, United States

*For correspondence: m-ashley-spies@uiowa.edu (MAS); maria-spies@uiowa.edu (MS)

Competing interests: The authors declare that no competing interests exist.

**Abstract** The DNA repair protein RAD52 is an emerging therapeutic target of high importance for BRCA-deficient tumors. Depletion of RAD52 is synthetically lethal with defects in tumor suppressors *BRCA1, BRCA2* and *PALB2*. RAD52 also participates in the recovery of the stalled replication forks. Anticipating that ssDNA binding activity underlies the RAD52 cellular functions, we carried out a high throughput screening campaign to identify compounds that disrupt the RAD52-ssDNA interaction. Lead compounds were confirmed as RAD52 inhibitors in biochemical assays. Computational analysis predicted that these inhibitors bind within the ssDNA-binding groove of the RAD52 oligomeric ring. The nature of the inhibitor-RAD52 complex was validated through an in silico screening campaign, culminating in the discovery of an additional RAD52 inhibitor. Cellular studies with our inhibitors showed that the RAD52-ssDNA interaction enables its function at stalled replication forks, and that the inhibition of RAD52-ssDNA binding acts additively with BRCA2 or MUS81 depletion in cell killing.

## Introduction

Understanding of synthetically lethal relationships between genome caretakers will help to define the molecular mechanisms underlying the maintenance of genomic integrity and may lead to the advancement of personalized cancer treatments. Depletion of the human DNA repair protein RAD52 is synthetically lethal with defects in tumor suppressors, BRCA1, BRCA2, or PALB2 (*Feng et al., 2011*; *Lok et al., 2012*; *Cramer-Morales et al., 2013*). Importantly, this synthetic lethality requires both copies of the tumor suppressor gene to be defective and should not manifest in the heterozygous cells. Therefore, specific RAD52 inhibitors are expected to selectively kill cancerous cells lacking one of these three tumor suppressors. Replacing or supplementing standard radiation and chemotherapies with the RAD52 inhibitors will help to decrease the toxicity associated with these treatments.

**eLife digest** Cells are constantly in danger of losing or scrambling critical genetic information because of DNA damage. To cope with this stress, cells have numerous DNA repair systems. One of these systems – homology-directed DNA repair – involves the proteins BRCA1 and BRCA2, which are often missing or defective in breast and ovarian cancers. The BRCA-deficient cancer cells can still survive, but become "addicted" to other DNA repair proteins – among them a protein called RAD52. It might be possible to kill these cancer cells using drugs that stop RAD52 from working. Such treatments would have the benefit of not harming normal healthy cells, as these cells contain working BRCA proteins and can survive without RAD52.

It is not currently known exactly how RAD52 allows the BRCA-deficient cells to survive, but this probably depends on RAD52's ability to bind to single strands of DNA. Small molecules that block the interaction between the RAD52 protein and DNA might therefore help to kill cancer cells.

Hengel et al. developed a high throughput biophysical method to search through a large collection of small molecules to find those that prevent RAD52 from binding to DNA. The best potential drug leads were then tested in laboratory-grown human cells and using biophysical and biochemical techniques. Computational approaches were also used to model how these molecules block the interaction between RAD52 and DNA at the atomistic level.

Hengel et al. then used the information about how the small molecules bind to RAD52 to perform further computational screening. This identified a natural compound that competes with single-stranded DNA to bind to RAD52. The activity of this molecule was then validated using biophysical methods.

The methods used by Hengel et al. provide the foundation for further searches for new anticancer drugs. Future studies that employ the small molecule drugs identified so far will also help to determine exactly how RAD52 works in human cells and how it helps cancer cells to survive.

BRCA1 and BRCA2 are tumor suppressors that are commonly mutated or depleted in hereditary and sporadic breast cancers, and have important roles in homologous recombination (HR) (*Prakash et al., 2015*), a template directed pathway that accurately repairs DNA lesions affecting both strands of the DNA duplex (*Couedel et al., 2004*; *Moynahan and Jasin, 2010*; *Jasin and Rothstein, 2013*; *Kowalczykowski, 2015*; *Heyer, 2015*). BRCA1 regulates repair pathway choice after DNA damage by promoting HR (*Kass and Jasin, 2010*; *Prakash et al., 2015*). BRCA2 is a recombination mediator, which facilitates assembly of the RAD51 nucleoprotein filament on ssDNA downstream of BRCA1 activities (*Couedel et al., 2004*; *Jensen et al., 2010*; *Liu et al., 2010*; *Thorslund et al., 2010*; *Prakash et al., 2015*). PALB2 mediates interactions between BRCA1 and BRCA2 proteins, acts as a scaffold connecting numerous tumor suppressors (*Park et al., 2014*), stimulates Polη-dependent DNA synthesis (*Buisson et al., 2014*) and RAD51 recombinase activity (*Dray et al., 2010*). Finally, the three tumor suppressors cooperate with Fanconi Anemia proteins in the repair of inter-strand DNA cross-links (*Kim and D'Andrea, 2012*).

Rad52 was identified in yeast as the main recombination mediator and the central player in the single-strand annealing pathway of mutagenic homology-directed DNA repair (*Mortensen et al., 2009*). In contrast to the severe recombination and repair phenotypes observed in yeast, deletion of *RAD52* has only a mild effect on recombination in vertebrates (*Rijkers et al., 1998*; *Yamaguchi-Iwai et al., 1998*; *Yanez and Porter, 2002*). Although it is clear that RAD52 is important for survival and uncontrolled proliferation of BRCA-deficient cancer cells, the molecular mechanism by which RAD52 allows BRCA-deficient cells to survive is unknown. The proposed mechanisms included the putative RAD52 recombination mediator function and its role in single-strand annealing pathway of homology-directed DSB repair (*Lok and Powell, 2012*). Functional interactions between BRCA1, BRCA2, PALB2 and RAD52, as well as the ability of RAD52 to promote BRCA-independent cell survival, are commonly expected to involve HR-related mechanisms. The recent discovery that BRCA proteins act together with the Fanconi Anemia pathway to support and protect replication forks points to a potentially more complex scenario (*Schlacher et al., 2011*; *2012*). Additionally, RAD52 cooperates with the structure-selective nuclease MUS81/EME1 to generate DNA double-strand

breaks (DSBs) essential for the recovery of stalled replication forks in the absence of the replication check point (*Murfuni et al., 2013*).

Known biochemical functions of human RAD52 include annealing of two complementary ssDNA strands in the presence of replication protein A (RPA) (*Van Dyck et al., 2001*; *Grimme et al., 2010*) and the ability to pair ssDNA to complementary homologous regions in supercoiled DNA (*Kagawa et al., 2001*; *Murfuni et al., 2013*). Putative recombination mediator activity of RAD52 (*Benson et al., 1998*) should also require ssDNA binding. Therefore, if the cellular functions of the RAD52 protein depend on the ssDNA binding, then inhibition of the RAD52-ssDNA interaction should have similar consequences as RAD52 depletion. RAD52 forms an oligomeric ring (*Kagawa et al., 2002*; *Lloyd et al., 2002*; *Singleton et al., 2002*; *Stasiak et al., 2000*), where the primary ssDNA binding site is located in the narrow groove spanning the ring circumference (*Lloyd et al., 2005*; *Mortensen et al., 2002*). We designated this ssDNA-binding groove as the feature to be targeted by small molecule inhibitors. While disrupting the protein-ssDNA interaction with small molecules presents a formidable challenge (*Yap et al., 2012*) that has only been overcome in a handful of cases, the ssDNA binding groove of RAD52 (for reasons discussed below) is a promising target and is distinct from the ssDNA binding sites of other ssDNA binding proteins.

Here, we report the development of a novel FRET-based high throuput screening (HTS) assay that led to the identification of compounds that disrupt the RAD52-ssDNA interaction. Initial HTS hits were biochemically validated in RAD52 functional assays and tested in two separate cellular assays. Two available high resolution crystal structures (PDB: 1H2I and 1KNO) of the conserved ssDNA-binding domain of RAD52 highlight the unique nature of this target (*Singleton et al., 2002*; *Kagawa et al., 2002*). The ssDNA-binding region is continuous around the circumference of the ring and has shallow sub-pockets that are repeating in each monomer. While the truncated version of RAD52 in the crystal structures may differ from the full length RAD52, it likely recapitulates the structural features of the ssDNA-binding groove. Computational docking followed by all atom-simulated annealing placed all identified RAD52 inhibitors into two distinct sub-pockets within the ssDNA-binding groove. Compounds '1' ((−)−Epigallocatechin) and '6' (Epigallocatechin-3-monogallate) predicted to bind within the RAD52 ssDNA-binding site, inhibited the formation of the RAD52-dependent DSBs in hydroxyurea (HU)-stressed, checkpoint deficient cells to the same level as RAD52 depletion. Moreover, '1' acts additively with the MUS81 depletion to kill cells treated with hydroxyurea (HU), which perturbs replication, and with checkpoint inhibitor UCN01. These data strongly suggest that the ssDNA binding activity of RAD52 is required for recovery of stalled replication forks in checkpoint deficient cells. We also show that '1' selectively kills cells depleted of BRCA2, further supporting the importance of the RAD52-ssDNA interaction in BRCA deficient cells and the potential therapeutic value of RAD52 inhibition. Finally, in order to validate the strength of our hypotheses about the structural nature of the RAD52-inhibitor complex, we developed a validated in silico screening campaign, based on our HTS results, using a library of four thousand natural products. We describe the discovery of **NP-004255**, a macrocyclic compound, which we show by NMR WaterLOGSY and biophysical assays to be a completely novel and effective inhibitor of the RAD52-ssDNA interaction. The implication of these findings for the discovery of novel therapeutics that specifically inhibit the activity of RAD52 is discussed.

## Results

### High throughput screening (HTS) of the MicroSource SPECTRUM collection identifies compounds that inhibit the RAD52-ssDNA interaction

To identify compounds that disrupt the RAD52-ssDNA interaction we adapted a previously developed FRET-based assay (*Grimme and Spies, 2011*; *Grimme et al., 2010*) to the HTS format. The RAD52-ssDNA interaction is independent of sequence and involves a binding site size of 4 nucleotides per monomer (*Singleton et al., 2002*). Our FRET-based assay relies on the ability of RAD52 to bind and wrap ssDNA around the narrow groove spanning the circumference of the protein ring (*Grimme and Spies, 2011*; *Grimme et al., 2010*). Förster Resonance Energy Transfer (FRET) donor (Cy3) and acceptor (Cy5) fluorophores are positioned at the ends of a 30-mer ssDNA (Cy3-dT$_{30}$-Cy5). When this ssDNA forms a stoichiometric complex with RAD52 (one 30-mer ssDNA molecule

per one heptameric ring of RAD52), the two fluorophores are brought close to one another resulting in an increase in the FRET signal. The assay was successfully adapted to the 384-well plates HTS format. In each well, we recorded the fluorescence signal of the Cy3 dye, which was excited directly, and the signal of Cy5 dye, which was excited via the energy transfer from Cy3. The apparent FRET signal was then calculated as described in the Materials and methods. The separation between the positive control (a stoichiometric complex of RAD52 with Cy3-dT$_{30}$-Cy5 substrate challenged with an excess of unlabeled ssDNA (Poly dT100)) and the negative control (an unperturbed stoichiometric complex of RAD52 with Cy3-dT$_{30}$-Cy5) initially resulted in a Z' factor of 0.66 when calculated for the whole plate. Further optimization increased the Z' factor calculated for the control rows in the screening experiments to 0.94, indicating excellent reliability of the assay (*Figure 1a*). Using this assay, we screened the MicroSource SPECTRUM collection, which contains 2320 drug and drug-like synthetic compounds as well as natural products, which represent a wide structural diversity and a range of known biological activities. The screening was carried out at 15 µM concentration of each compound in the library. Of the 2320 compounds examined, 96 were identified as preliminary hits. The results for a one plate in the collection are shown in *Figure 1b* with initial hits that were validated in the follow up experiments highlighted in green. These preliminary hits were selected based on the criterion of their separation from the negative control (RAD52 + Cy3-dT$_{30}$-Cy5) of at least 5 S.D. The 96 preliminary hits were assembled into a 'cherry picked plate' and were tested in two more rounds of screening *Figure 1c*. Compounds that showed reproducible and nearly complete inhibition were re-tested at a range of small-molecule concentrations. Seven of the compounds tested showed a promising decrease in FRET in the re-screening assays and six were selected for biochemical validation (shown in green). One compound was excluded due to a low molecular weight and promiscuous binding observed in the follow-up biochemical assays. Additionally, we selected six compounds that elicited the FRET values below the positive control. These molecules were expected either to be 'false positives' (i.e. molecules that are fluorescent in the Cy3 channel or interact with DNA) or to have a significant absorbance in the region of Cy3 emission and/or Cy5 excitation. We purchased 12 compounds and confirmed their chemical structures by 1D NMR. The identified compounds and their chemical structures are listed in the *Table 1*.

## Selected compounds inhibit ssDNA binding and wrapping by RAD52 with IC$_{50}$ values ranging from mid-nanomolar to high-micromolar range

In order to determine how the selected compounds affect known RAD52 functions we performed FRET-based assays that recapitulate the HTS screen, yet have higher precision and yield a calibrated FRET signal. We titrated increasing amounts of each compound into a cuvette containing preformed stoichiometric RAD52-Cy3-dT$_{30}$-Cy5 complexes (1 nM Cy3-dT$_{30}$-Cy5 and 8 nM RAD52). As the compounds bind and disrupt the ssDNA-RAD52 interaction we observed a decrease in FRET between the DNA-tethered Cy3 and Cy5 fluorophores. At each concentration of the compound, the FRET signal was adjusted for the change in Cy3 and Cy5 fluorescence in the presence of the compound, but in the absence of protein. From the hyperbolic inhibition curves we then calculated IC$_{50}$ value for each compound under these conditions (*Figure 2b* and *3b*, *Table 1*). IC$_{50}$ values were in the nanomolar range for compounds '5', '6', '14', '15' and '19'. We calculated IC$_{50}$ values in the micro-molar range for compounds '1', '3', '7', '13', and '16'. Compound '4' had IC$_{50}$ value in the mid micromolar range. Compounds '17' and '18' were poor inhibitors of ssDNA binding with IC$_{50}$ values in the high micro-molar range. These compounds were likely false positives in our HTS screen.

## Selection of the compounds that inhibit RAD52-mediated ssDNA annealing

To determine how the selected compounds affect the ssDNA annealing function of RAD52 we performed FRET-based annealing assays (*Grimme et al., 2010*; *Grimme and Spies, 2011*). These assays utilize two complementary single stranded 28-nucleotide-long substrates, which contain either Cy3 (T-28) or Cy5 (P-28) incorporated into the middle of the respective DNA strand. When the substrates are annealed by RAD52, the Cy3 and Cy5 dyes are separated by 3 base pairs and yield a high FRET signal (*Figure 2—figure supplement 2*). Negative controls containing T-28 and P-28 with the compounds in the absence of RAD52 displayed no change in FRET suggesting the small molecules do not promote ssDNA annealing by themselves. The annealing reactions were initiated by mixing two

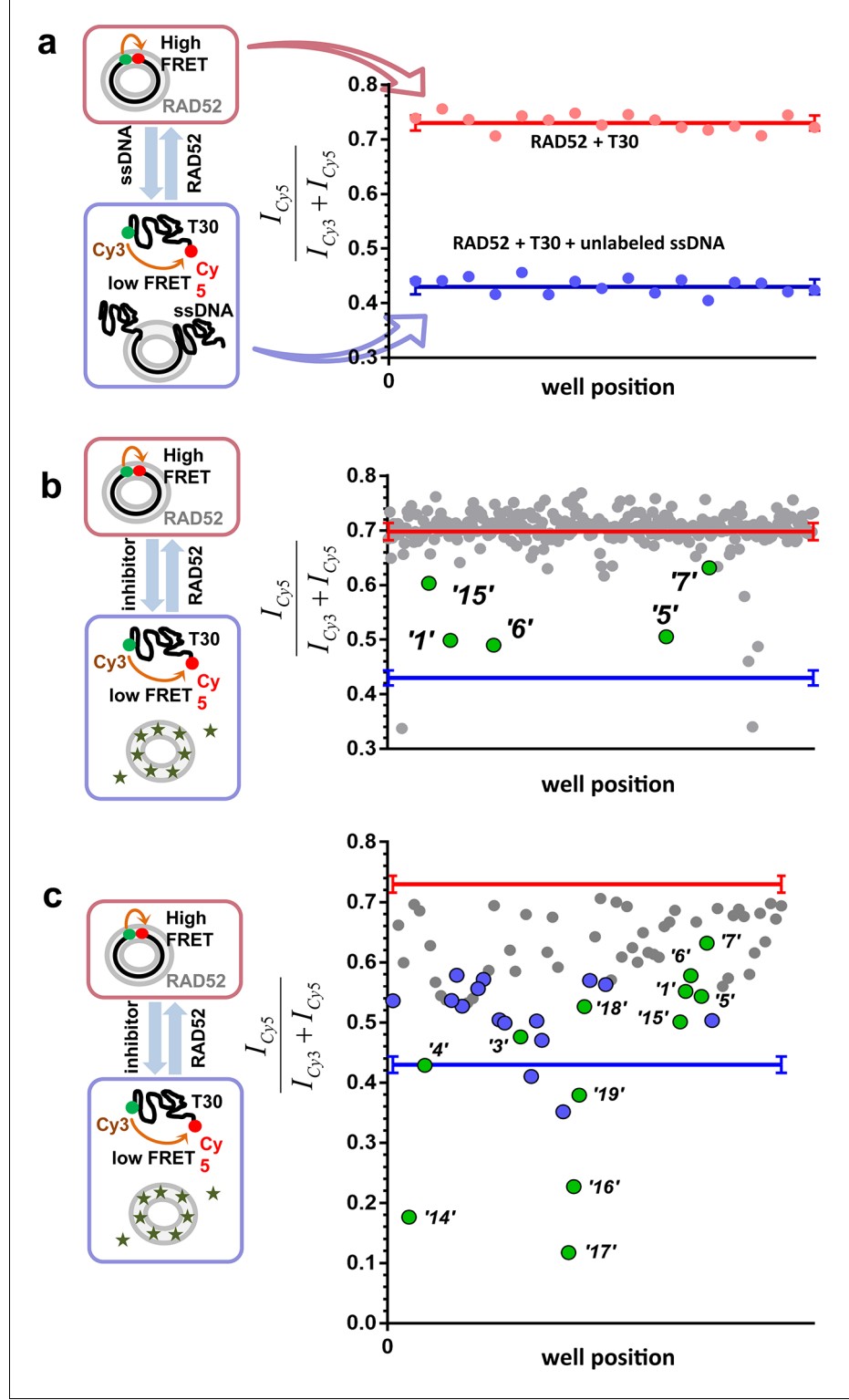

**Figure 1.** High throughput screening of the MicroSource SPECTRUM collection identifies 12 compounds that inhibit the RAD52-ssDNA interaction. (a) Control lanes from a 384 well: 16 negative control wells contain stoichiometric RAD52- Cy3-dT$_{30}$-Cy5 complexes (red filled circles), while 16 positive control wells contain a stoichiometric RAD52- Cy3-dT$_{30}$-Cy5 complex challenged with unlabeled polydT100 (blue filled circles). Red and blue lines with error bars at the ends indicate the average and the standard deviation for the negative and positive controls, respectively. Z' factor of 0.94 was calculated for these control lanes, indicating excellent reliability of the

*Figure 1 continued on next page*

*Figure 1 continued*

assay. (b) A representative 384 well plate from the HTS screen highlighting '*1*', '*5*', '*6*', '*7*', and '*15*'. Red and blue lines with error bars at the ends indicate the average and the standard deviation for the negative and positive controls, respectively. (c) Average of cherry-picked rescreening of compounds identified from screening all plates in the MicroSource SPECTRUM collection highlighting all 12 identified hits (green filled circles) along with a number of false positive compounds (blue filled circles) that either showed poor reproducibility in subsequent rescreening or a linear dependence of the signal on the compound concentration. Red and blue lines with error bars at the ends indicate the average and the standard deviation for the negative and positive controls, respectively.

half reactions and observing the change in FRET over time in the presence of varying concentrations of small molecules. An increasing FRET value over time indicates the formation of the dsDNA duplex which brings the two dyes in close proximity (*Figure 2—figure supplement 2*). Fitting the annealing data to a double exponential allowed us to calculate and compare the final extent of annealing at varying concentrations of each compound compared to RAD52 alone. We plotted the final extent of annealing vs the concentration of the compound and calculated an $IC_{50}$ of annealing inhibition. A full set of the annealing time courses recorded at different concentrations of '*1*' is shown in *Figure 2—figure supplement 2*. As the concentration of the compound increases, the final extent of annealing is reduced compared to RAD52 alone. Since we showed previously that ssDNA wrapping around the RAD52 ring is necessary for the most efficient annealing (*Grimme et al., 2010*; *Honda et al., 2011*), it was expected that a compound that interferes with ssDNA access to the ssDNA binding groove of RAD52 would compete with ssDNA annealing, thus shifting the equilibrium away from the dsDNA product. It is notable that the $IC_{50}$ values for DNA annealing were generally higher than $IC_{50}$ values for the ssDNA binding. We attribute this to the dynamic nature of the RAD52-ssDNA complex as well as to RAD52 ability to bypass regions of heterology and other obstacles during the homology search process (*Rothenberg et al., 2008*). To confirm specificity of the two compounds ('*1*' and '*6*') selected for the in-depth follow-up characterization as disruptors of the RAD52-ssDNA interaction, we tested the ability of these compounds to interfere with the RAD52-dsDNA interaction, which involves a different site on the RAD52 ring (*Kagawa et al., 2008*; *Grimme et al., 2010*). At the stoichiometric RAD52: dsDNA ratio, (1 nM dsDNA: 10 nM RAD52) the dsDNA is bent upon RAD52 binding, which allows us to distinguish the RAD52-bound and free dsDNA (*Figure 2—figure supplement 1*). Interestingly, '*1*' had no effect on the RAD52-dsDNA interaction (*Figure 2b* open grey squares), which indirectly confirms its specificity for the ssDNA-binding groove of RAD52. In contrast, '*6*' was able to displace dsDNA from the RAD52-dsDNA complex (*Figure 3b* open grey squares). Dynamic light scattering experiments conducted in the presence of equimolar concentrations of each compound and RAD52 showed that the presence of these compounds neither breaks up the oligomeric ring of RAD52 nor causes protein aggregation (*Figure 2—figure supplement 3*). Notably, this means that our compounds act differently from the RAD52 inhibitor 6-hydroxy-DL-dopa (*Chandramouly et al., 2015*), which disrupts supramolecular assembly of the RAD52 protein. We further confirmed that the inhibition of the ssDNA binding does not occur due to aggregation of compounds as annealing FRET trajectories in the presence of 0.01% Triton X-100 are identical to those in the absence of Triton X-100.

## Compounds '1' and '6' physically interact with RAD52

To confirm that the selected compounds bind RAD52, we employed water-ligand observation with gradient spectroscopy (WaterLOGSY), an NMR technique, which is based on transfer of magnetization from bulk water to the protein-bound compound of interest (*Dalvit et al., 2001*, *2000*). In WaterLOGSY spectrum, if a compound binds to a protein, the compound will receive negative nuclear Overhauser effects (NOEs) due to the slow tumbling of the protein-compound complex, leading to a positive WaterLOGSY peak. In contrast, if a compound does not bind to a protein, the compound will receive positive NOEs due to the fast tumbling of the compound itself, resulting in a negative WaterLOGSY peak. *Figure 2a* and *Figure 3a* show that both '*1*' and '*6*' physically interact with RAD52 protein. When the aromatic region of the 1D $^1$H NMR spectrum of compound '*1*' alone (black) and the WaterLOGSY spectrum of compound '*1*' in the presence of RAD52 (red) are

**Table 1.** The twelve hits from the FRET-based HTS assay aimed at finding inhibitors of the RAD52-ssDNA interaction.

| # | Small molecule name; CAS # | Small molecule structure | IC$_{50}$ (DNA binding); FRET value at saturation | IC$_{50}$ (Annealing extent) | SAEM ΔG (kcal/mol) |
|---|---|---|---|---|---|
| '1' | (−)−Epigallocatechin; 970-74-1 | | ssDNA: 1.8 ± 0.1 µM; 0.45 ± 0.004 ssDNA-RPA: 1.6 ± 0.1 µM; | ssDNA: 4.9 ± 0.4 µM ssDNA-RPA 4.8 ± 1.8 µM; | −8.60 |
| '3' | Methacycline Hydrochloride; 3963-95-9 | | 2.0 ± 0.17 µM; 0.47 ± 0.01 | 3.8 ± 0.2 µM | −4.61 |
| '4' | Rolitetracycline; 751-97-3 | | 29 ± 8.2 µM; 0.56 ± 0.04 | NI | −10.5 |
| '5' | (−)−Epicatechin gallate; 1257-08-5 | | 255 ± 16 nM; 0.41 ± 0.004 | 20 ± 0.7 µM | −9.87 |
| '6' | Epigallocatechin-3-monogallate; 989-51-5 | | ssDNA: 277 ± 22 nM; 0.46 ± 0.01 ssDNA-RPA: 1.6 ± 0.5 µM; | ssDNA: 6.7 ± 2.1 µM ssDNA-RPA: 3.7 ± 0.5 µM; | −10.69 |
| '7' | (−)−Epicatechin; 490-46-0 | | 1.45 ± 0.11 µM; 0.51 ± 0.01 | NI | −9.03 |
| '14' | Oxidopamine; 28094-15-7 1199-18-4 | | 779 ± 51 nM; 0.50 ± 0.01 | NI | −5.71 |
| '15' | Quinalizarin; 81-61-8 | | 563 ± 40 nM; 0.51 ± 0.01 | 5.6 ± 0.6 µM | −9.17 |
| '16' | Cisapride Monohydrate; 260779-88-2 81098-60-4 | | 1.06 ± 0.05 µM; 0.50 ± 0.01 | NI | −8.39 |
| '17' | Cedrelone; 1254-85-9 | | >300 µM | NI | −10.0 |
| '18' | Asiatic Acid; 464-92-6 18449-41-7 | | >800 µM | >100 µM | −11.33 |
| '19' | Gossypetin; 489-35-0 | | 913 ± 58 nM; 0.49 ± 0.01 | 6.0 ± 2.3 µM | −9.30 |

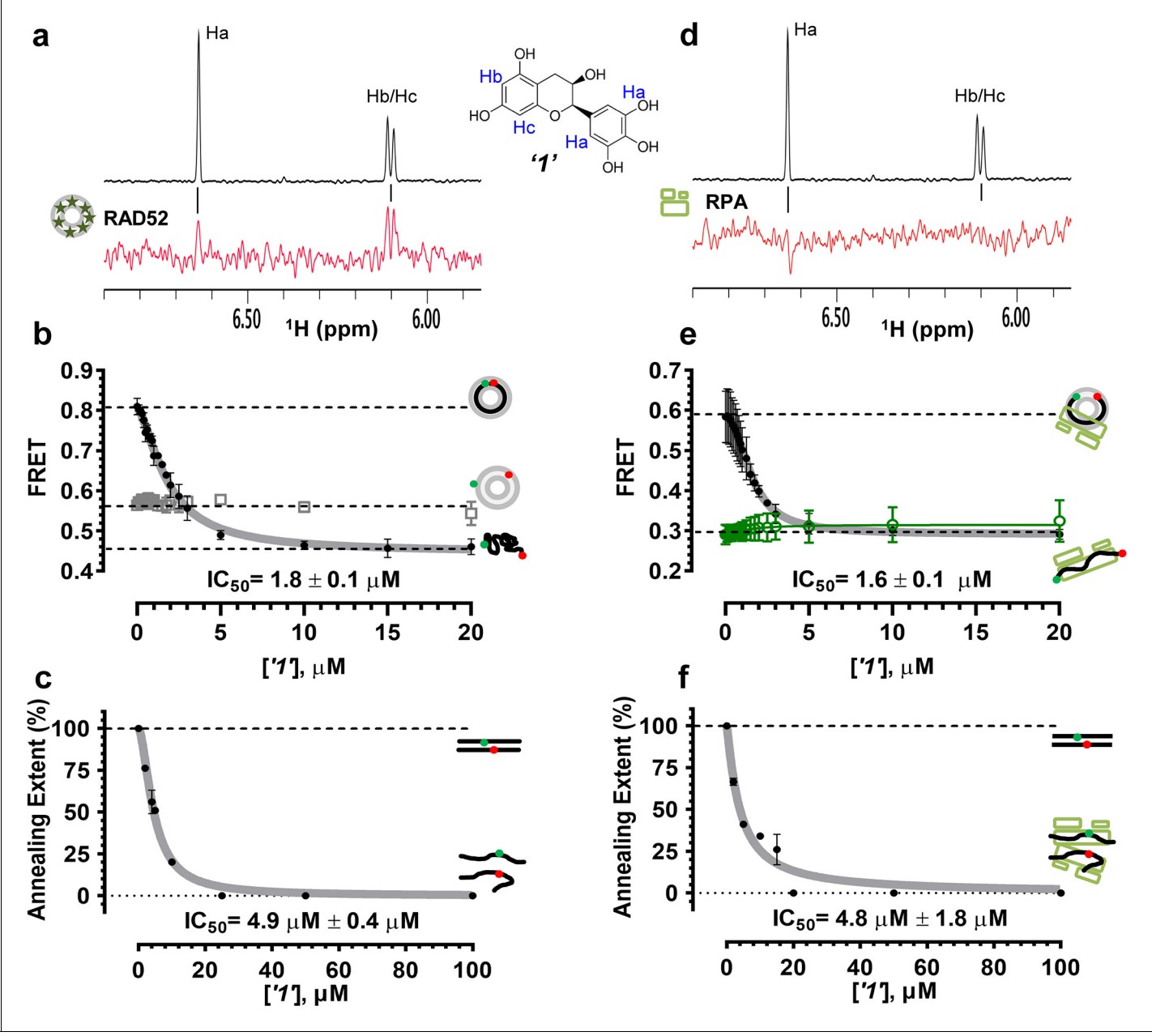

**Figure 2.** Biochemical characterization of '1'. (a) Aromatic region of the 1D $^1$H NMR spectrum of compound '1' alone (black) and the WaterLOGSY spectrum of 20 µM compound '1' in the presence of 3.3 µM RAD52 (red). The nonexchangeable proton peaks are labeled using atom names as indicated on the structure of compound '1'. (b) IC$_{50}$ values for inhibition of ssDNA binding and wrapping were determined using FRET-based assays that follow the change in geometry of a Cy3-dT30-Cy5 substrate (black circles). The computed IC$_{50}$ value is shown below the curve. Titration of the RAD52-dsDNA with '1' (grey boxes) shows no perturbation of the dsDNA binding. (c) IC$_{50}$ values for inhibition of RAD52-mediated ssDNA annealing were determined by fitting the dependence of the extent of oligonucleotide-based annealing reaction carried in the presence of increasing concentration of '1'. (d) Aromatic region of the 1D $^1$H NMR spectrum of compound '1' alone (black) and the WaterLOGSY spectrum of 20 µM compound '1' in the presence of 3.3 µM RPA (red). (e) Titration of the RAD52-RPA- Cy3-dT$_{30}$-Cy5 complex with '1' (black circles). The computed IC$_{50}$ value is shown below the curve. Green squares show titration of the RPA- Cy3-dT$_{30}$-Cy5 complex with '1'. (f) IC$_{50}$ values for inhibition of RAD52-mediated annealing of the RPA-coated ssDNA were determined by fitting the dependence of the extent of the annealing reaction carried out in the presence of increasing concentration of '1'.

The following figure supplements are available for figure 2:

**Figure supplement 1.** Stoichiometric complexes of RAD52 with ssDNA, RPA-coated ssDNA and dsDNA yield characteristic FRET values.

*Figure 2 continued on next page*

*Figure 2 continued*

**Figure supplement 2.** RAD52 FRET based ssDNA annealing assay in the presence of small molecules.

**Figure supplement 3.** None of the tested compounds affect the oligomeric state of RAD52 protein.

**Figure supplement 4.** Compounds '1' and '6' have no effect on the interaction between RAD52 and RPA proteins.

compared, positive WaterLOGSY peaks are observed for the compound '*1*', indicating the binding of '*1*' to RAD52 (*Figure 2a*). Similarly, the binding of '*6*' to RAD52 is evident from the positive WaterLOGSY peaks that are clearly detected for this compound (*Figure 3a*). Notably, '*6*' also binds to RPA as shown by the positive WaterLOGSY peaks (*Figure 3d*), thought it does not interfere with the RPA-ssDNA interaction (*Figure 3e*), while '*1*' neither binds to RPA as shown by the negative WaterLOGSY peak (*Figure 2d*) nor interferes with the RPA-ssDNA interaction (*Figure 2e*).

## Compounds '1' and '6' inhibit RAD52 binding to and annealing of the RPA-coated ssDNA

In the cell, ssDNA is typically found in complex with Replication protein A (RPA), the major eukaryotic ssDNA-binding protein essential for DNA replication, repair and recombination (*Wold, 1997*; *Oakley and Patrick, 2010*; *Chen and Wold, 2014*). The RPA-ssDNA complex is a physiologically relevant substrate for the RAD52-mediated strand annealing. To confirm that compounds '*1*' and '*6*' can inhibit the RAD52 binding to and annealing of the RPA-coated ssDNA we added stoichiometric amounts of RPA (1 RPA per 30 nucleotides of ssDNA) to the FRET-based ssDNA binding/wrapping and ssDNA annealing experiments described above. RPA binds ssDNA with high affinity and extends the ssDNA to its contour length. In our assays such an extension manifests as a distinct FRET state of ~0.3, which is readily distinguished from a FRET state of ~0.48 of free Cy3-dT$_{30}$-Cy5 ssDNA, as well as ~0.63 FRET of the stoichiometric ssDNA-RPA-RAD52 complex (see *Figure 2—figure supplement 1* and (*Grimme and Spies, 2011*) for details). Notably, neither '*1*' nor '*6*' affected the RPA-ssDNA interaction over the range of the tested compound concentrations (*Figure 2e* and *Figure 3e*). Both, however, inhibited the RAD52 binding to and wrapping of RPA-coated ssDNA with IC$_{50}$ values identical to those determined without RPA (*Figure 2e* and *Figure 3e*). Similarly, we confirmed that both '*1*' and '*6*' inhibit the RAD52-mediated annealing of RPA-coated ssDNA with the IC$_{50}$ values comparable to the inhibition of ssDNA annealing (*Figure 2f* and *Figure 3f*). Notably, this inhibition is not due to the disruption of the RAD52-RPA interaction as neither '*1*' nor '*6*' interfered with the interaction between the two proteins (*Figure 2—figure supplement 4*).

## Virtual screening places the RAD52 inhibitors within the ssDNA binding groove

In order to gain insight into the binding determinants of the various polyphenol hits obtained from the HTS screening, we undertook a computational investigation using the structure of the oligomeric ring formed by the conserved ssDNA-binding domain of RAD52 (PDB 1KNO). We utilized a layered approach involving docking and all atom-simulated annealing with explicit solvent, using a knowledge based force field (*Krieger et al., 2004*). The long circular ssDNA binding groove of the RAD52 oligomeric ring yielded excellent 'druggability' scores (~4.0), based on the pocket metric of Sugo et al., (*Soga et al., 2007*). Docking approaches generally generate many potential poses, and many false positives. Initially, top scoring poses in either Triangle Matcher (placement), London dG (affinity scoring function) or MM/GBSA (physics based scoring) were retained for further analysis (see Materials and methods section for details).

An all atom force field-based protocol was employed to distinguish binding affinities from a variety of docking poses that possessed various docking metrics. We compared the different scoring metrics, such as Triangle Matcher placement scores followed by rescoring with the affinity function versus the computationally expensive force field-based ligand refinement and subsequent MM/GBSA scoring. The use of all atom simulations (including explicit solvent models), when combined with docking, has been shown to significantly boost docking procedures' ability to predict and rank

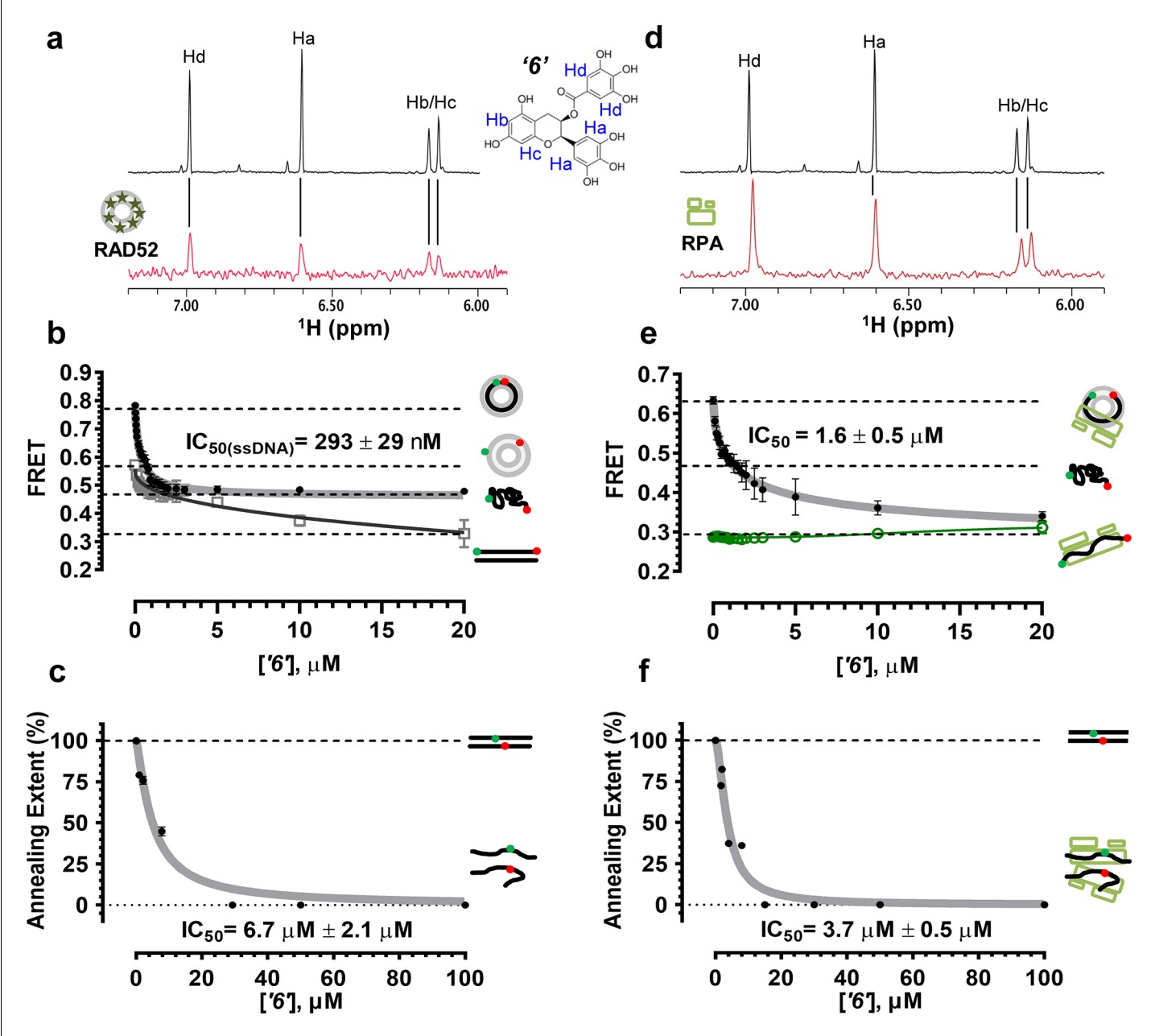

**Figure 3.** Biochemical characterization of '6'. (a) Aromatic region of the 1D $^1$H NMR spectrum of compound '6' alone (black) and the WaterLOGSY spectrum of 40 µM compound '6' in the presence of 3.3 µM RAD52 (red). The nonexchangeable proton peaks are labeled using atom names as indicated on the structure of compound '6'. (b) IC$_{50}$ values for inhibition of ssDNA binding and wrapping were determined using FRET-based assays that follow the change in geometry of a Cy3-dT30-Cy5 substrate (black circles). The computed IC$_{50}$ value is shown above the curve. Titration of the RAD52-dsDNA with '6' (grey boxes) shows that this inhibitor also perturbs the RAD52-dsDNA interaction. (c) IC$_{50}$ values for inhibition of RAD52-mediated ssDNA annealing were determined by fitting the dependence of the extent of oligonucleotide-based annealing reaction carried in the presence of increasing concentration of '6'. (d) Aromatic region of the 1D $^1$H NMR spectrum of compound '6' alone (black) and the WaterLOGSY spectrum of 40 µM compound '6' in the presence of 3.3 µM RPA (red). (e) Titration of the RAD52-RPA- Cy3-dT$_{30}$-Cy5 complex with '6' (black circles). The computed IC$_{50}$ value is shown below the curve. Green squares show titration of the RPA- Cy3-dT$_{30}$-Cy5 complex with '6'. (f) IC$_{50}$ values for inhibition of RAD52-mediated annealing of the RPA-coated ssDNA were determined by fitting the dependence of the extent of the annealing reaction carried out in the presence of increasing concentration of '6'.

compound affinities. Therefore, we can compare the differences in free energy changes due to ligand binding for each of the poses, an ability that is not all within the realm of classical docking procedures (*Ellingson et al., 2015*; *Whalen et al., 2011*; *Warren et al., 2006*; *Head, 2010*). Thirty four unique docking poses were selected for comparison for each compound. Comparisons of the results of the various scoring metrics for docking of ligands to the RAD52 complex showed a clear lack of consensus between the three methods, except for compound '*6*', which resulted in a single pose scoring the highest in all three methods. To determine which methodology consistently provides the most accurate scoring, we employed a conservative approach, in which the 34 selected top scoring complexes were further subjected to all atom-simulated annealing studies, using explicit solvent and salt conditions.

The binding affinities computationally determined using the Simulated Annealing Energy Minimization (SAEM) Docking approach are listed in the *Table 1*). Interestingly, in most cases, the RAD52-ligand complex resulting from the best placed docking pose, rather than the more computationally intensive MM/GBSA (physics-based) scoring function, yielded the final SAEM-generated RAD52-ligand complex with the lowest energy. Compounds '*1*' and '*6*' yielded complexes with unique binding sub-pockets or 'hotspots' along the RAD52 binding groove, suggesting that they may have distinct biological activities and/or efficacies with regard to their ability to compete with ssDNA binding (*Figure 4*). The SAEM-generated complexes indicate that compounds '*1*' and '*6*' occupy complex pockets lying at the interface of two RAD52 monomers. Notably, all final compound placements include interactions, directly or through the interstitial water molecules, with key RAD52 residues, which have been previously shown to be involved in ssDNA binding (*Lloyd et al., 2005*) (*Figure 4*). In particular, R55, Y65, K152, R153 and R156 found in the vicinity of the docked compounds (*Figure 4b*) have been shown to impact ssDNA binding (*Lloyd et al., 2005*). Additional participants in the binding of our inhibitors include K141 and K144 residues that are important to distinct cellular functions of yeast Rad52. A highly conserved K144 corresponds to K159 in *S. cerevisiae* Rad52. Its K159A substitution results in severe deficiency in mitotic recombination, mild γ-ray sensitivity, but unperturbed recombination between direct repeats (*Mortensen et al., 2002*). K141 corresponds to *S. cerevisiae* R156, whose substitution to alanine causes γ-ray sensitivity only (*Mortensen et al., 2002*).

All of the tested inhibitors yielded complexes in which interstitial solvent plays a role in the binding of the ligand. Unlike classic enzyme pockets, which often have large desolvated volumes, the RAD52 ssDNA binding groove cannot truly be evaluated for the ability to bind to compounds without understanding the role of solvent in its various sub-pockets. The SAEM method used here was specifically designed to capture these complex recognition parameters. Unlike the case of deeply buried waters that occur in many active site pockets of enzymes, the waters along the RAD52 ssDNA-binding groove are mostly not involved in productive interstitial H-bonding with the ligand, but rather, represent a van der Waals binding surface, suggesting opportunities for future ligand improvement.

## Inhibiting the RAD52-ssDNA interaction interferes with RAD52/MUS81-mediated DSB formation essential for the replication fork recovery in check point deficient cells

In human cells, RAD52 may perform both limited recombination-mediator function in the RAD51-dependent pathway (*Lok and Powell, 2012*; *Benson et al., 1998*; *Feng et al., 2011*) as well as additional RAD51-independent roles (*Lok and Powell, 2012*; *Murfuni et al., 2013*; *McIlwraith and West, 2008*). One of these HR independent roles of RAD52 involves stimulation of MUS81/EME1-dependent DSB formation at replication forks stalled by hydroxyurea (HU) treatment in the absence of cellular checkpoints (*Murfuni et al., 2013*). Since the most likely targets of these MUS81/EME1/RAD52-dependent DSBs are the DNA structures produced by RAD52 (*Murfuni et al., 2013*), we expected this activity to depend on the RAD52-ssDNA interaction (*Figure 5a*). To confirm this, we assessed DSB formation in checkpoint deficient cells using the neutral comet assay. These assays monitor MUS81/EME1/RAD52-dependent DSB formation upon induction of replication stress by HU treatment in primary fibroblasts immortalized by hTERT expression and treated with UNC01 to inhibit CHK1 kinase (*Figure 5b*). Our data show that increasing amounts of '*1*' and '*6*' decrease the mean tail moment indicative of the decrease in the MUS81/EME1/RAD52-dependent DSB formation (*Figure 5*). Importantly, these inhibitors recapitulate RAD52 depletion by inhibiting RAD52-MUS81-

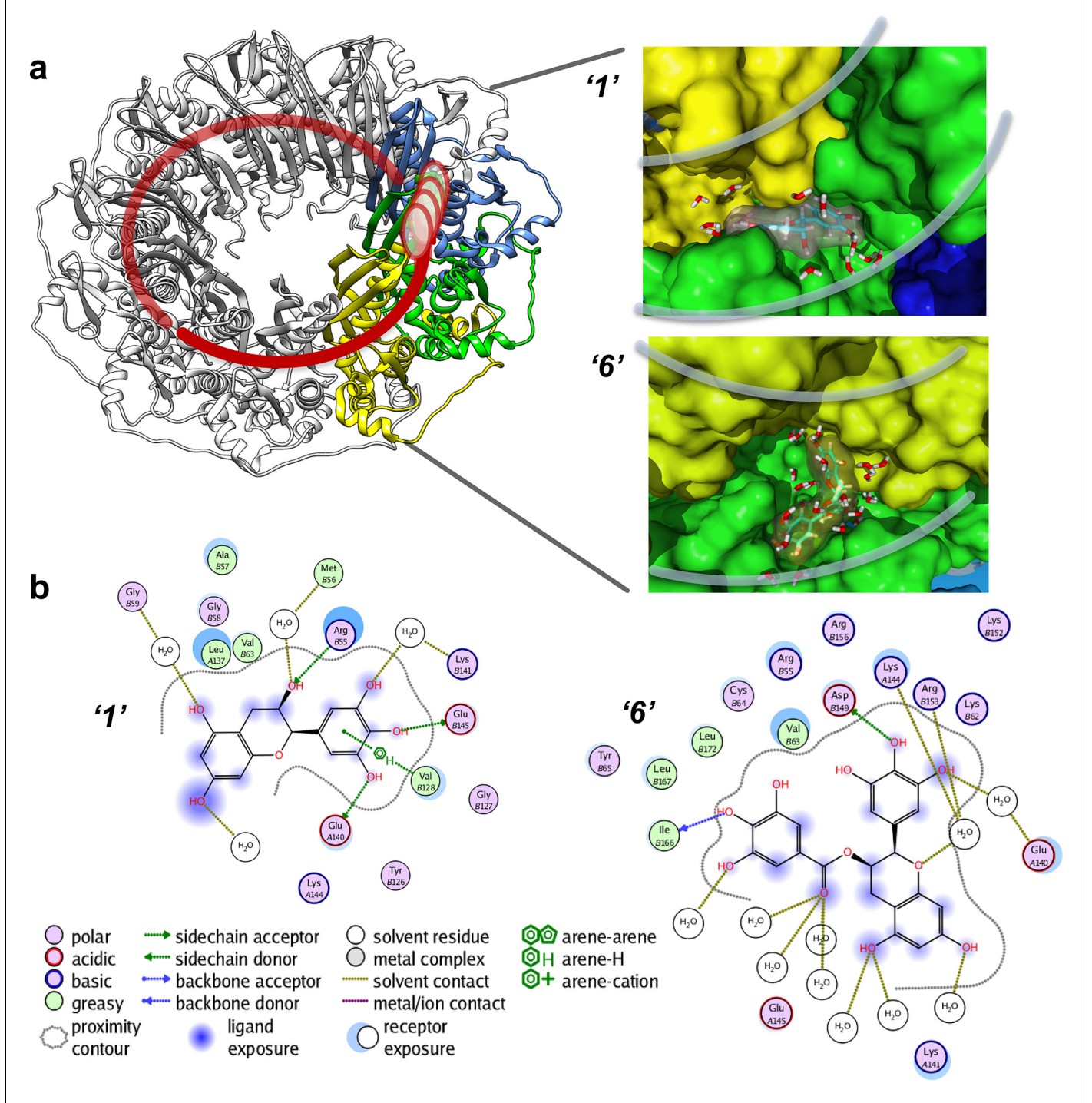

**Figure 4.** Virtual screening places the RAD52 inhibitors within the ssDNA binding groove. (**a**) Three individual monomers of the RAD52-NTD undecameric ring (PDB 1KNO) are colored yellow, green and blue respectively. '*1*' and '*6*' occupy similar sites at the interface of two subunits. Two grey lines in each panel indicate the approximate boundaries of the ssDNA-binding groove. (**b**). MOE ligand maps highlight water mediated interactions as well as interactions with amino acids. '*1*' likely mediates interactions through R55, V128, E140, and E145, as well as through water contacts made with G59, M56, and K141. '*6*' likely binds via hydrogen bonding via D149 and I166 as well as through water interactions with E140, K144, and R153.

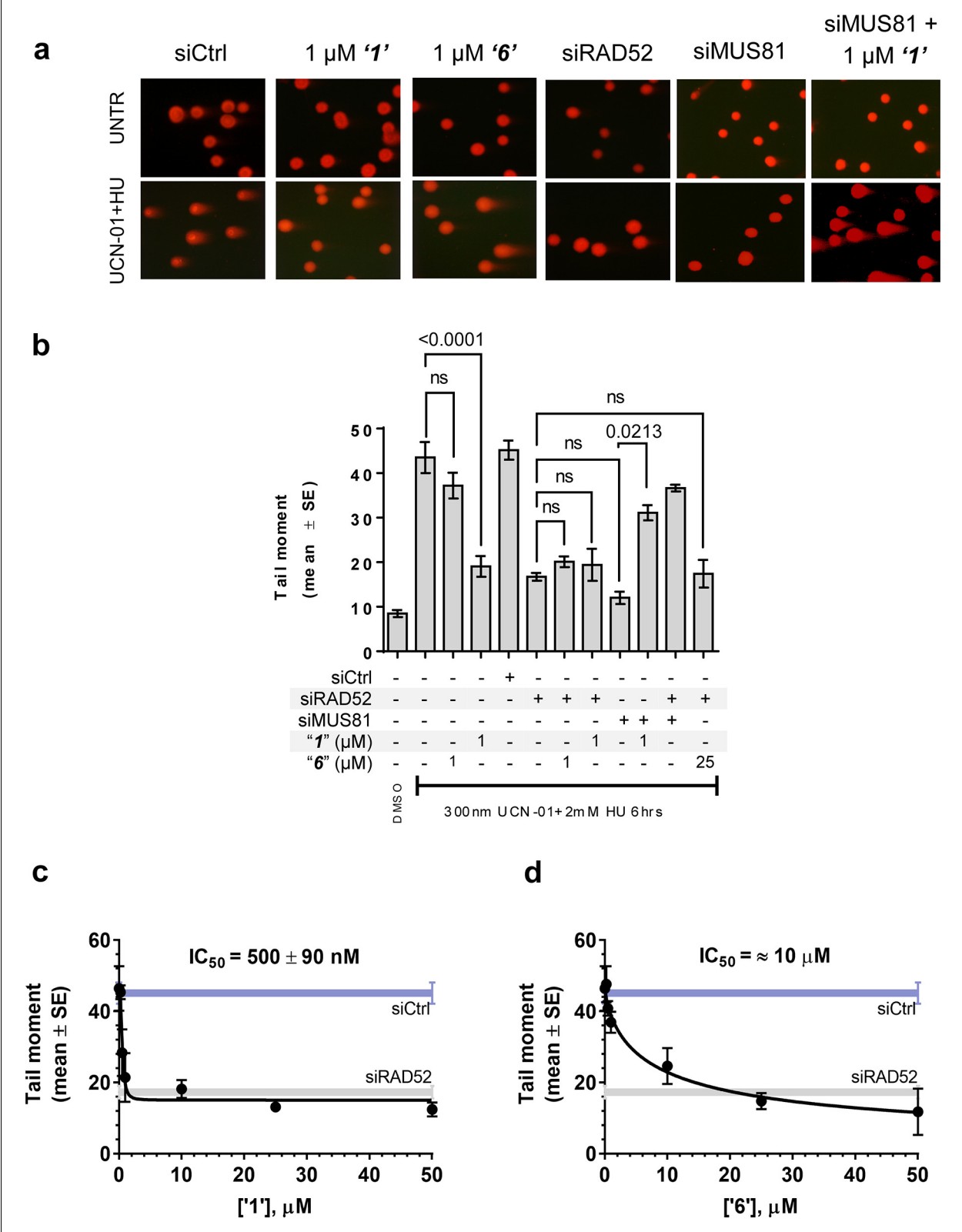

**Figure 5.** Inhibiting the RAD52-ssDNA interaction interferes with RAD52/MUS81-mediated DSB formation essential for replication fork recovery in check point deficient cells. (a) Representative images showing fields of cells from the comet assay for untreated, as well as from UCN01 (300 nM) and HU (2 mM) treated cells in the presence and absence of '1', '6', and siRAD52. (b) '1' and '6' at 1 or 25 µM recapitulate RAD52 depletion. GM1604 cells,

*Figure 5 continued on next page*

*Figure 5 continued*

transfected or not with siRNAs against RAD52, were treated as indicated, in the presence or absence of the inhibitor. At the end of treatment, DSBs were analyzed by neutral comet assay. Data are presented as the mean ± SEM from two independent experiments; p values are shown in the graph when differences are statistically significant. (c–d) Inhibitors '1' and '6', respectively decrease the mean tail moment following HU treatment with $IC_{50}$ values ranging from mid nanomolar to low micromolar. Cells were treated as in (b). Data are from three independent experiments.

The following figure supplement is available for figure 5:

**Figure supplement 1.** Compound '1' does not affect MUS81 activity.

dependent DSBs at stalled replication forks (*Figure 5a–b*). Notably, even at 500 nM, '1' had the same effect of reduction in DSBs as siRNA depletion of RAD52. These data strongly support the idea that inhibiting the RAD52-ssDNA interaction in cells recapitulates the effects of RAD52 depletion with respect to its role at the distressed replication forks. It also confirms our previous supposition that the target of MUS81/EME1-mediated cleavage under these conditions are indeed the structures annealed by RAD52. Interestingly, the concentrations of '1' sufficient to inhibit the MUS81/EME1/RAD52-mediated DSBs correlate well with the $IC_{50}$ values for inhibition of ssDNA binding/wrapping in vitro (compare *Figure 5c* with *Figure 2b and e*). These values are significantly lower than those required for inhibiting annealing of short, complementary oligonucleotides (*Figure 2c and f*). Higher concentration of '6' required to inhibit MUS81/EME1/RAD52-mediated DSBs (*Figure 5d*) is likely due to the particular chemical nature of this compound, which is a promiscuous binder; not only does it interact with RPA and binds within the dsDNA binding site of RAD52, but has been identified as an inhibitor in 192 different HTS assays (*Pubchem*). We previously reported that concomitant depletion of RAD52 and MUS81 gives raise to the MUS81-independent DSBs (*Murfuni et al., 2013*). In agreement with a specific activity towards RAD52, treatment of the MUS81-depleted cells with "1" resulted in an appearance of the MUS81-independent DSBs upon replication stress induced by CHK1 inhibition. Due to its lower capacity to inhibit the MUS81/EME1/RAD52-mediated DSBs, and its expected off-target effects, we have eliminated '6' from further analysis and focused all our subsequent cellular studies on '1', which appeared more specific in our biochemical studies and had no Pubchem hits. The fact that none of the compounds we tested showed additive effects on DSBs with RAD52 siRNA depletion, suggests that the effect of these inhibitors is specific to RAD52 at least with respect to recovery from replication stress. Furthermore, accumulation of anaphase bridges, a phenotype associated with impairment of the RAD52-independent mitotic function of MUS81/EME1 was completely unaffected by inhibition of RAD52, whereas it was strongly stimulated by MUS81 silencing (*Figure 5—figure supplement 1*). This observation strongly suggests that the suppression of the DSB formation is not due to direct inhibition of MUS81, but is mediated through RAD52 inhibition.

## Inhibiting the RAD52-ssDNA interaction kills BRCA2-depleted cells, as well as MUS81-depleted cells under pathological replication conditions

Compounds '1' and '6' are able to interfere with MUS81-dependent DSBs formation under pathological replication, mimicking RAD52 depletion (*Figure 5*). To address whether inhibition of the RAD52-ssDNA binding reduces viability of MUS81-depleted cells, as reported for RAD52 depletion (*Murfuni et al., 2013*), we evaluated cell death after inducing replication stress by pharmacological CHK1 inhibition (*Figure 6*). In cells transfected with the control (Ctrl) siRNAs, replication stress induced by HU treatment resulted in 20% cell death, which was increased similarly by MUS81 or BRCA2 knock-down by nearly two-fold. Treatment with '1' also potentiated the effect of the combined HU+UCN01 treatment and, strikingly enhanced cell death observed in MUS81-depleted cells. Inhibition of RAD52 increased cell death of MUS81-depleted cells even under unperturbed cell growth to approximately the same level as RAD52 depletion by siRNA (*Murfuni et al., 2013*). Strikingly, no additive effect on cell death was detected in cells depleted of RAD52 and treated with the RAD52 inhibitor, as compared with the cells transfected with the RAD52 siRNAs alone (*Figure 6b*).

Depletion of RAD52 not only enhances cell death of MUS81-depleted cells, but also reduces viability of BRCA2-deficient cells, making RAD52 an attractive target for potential treatment of BRCA2-deficient tumors (*Feng et al., 2011*; *Warren et al., 2006*). Treatment with '1' acted additively with

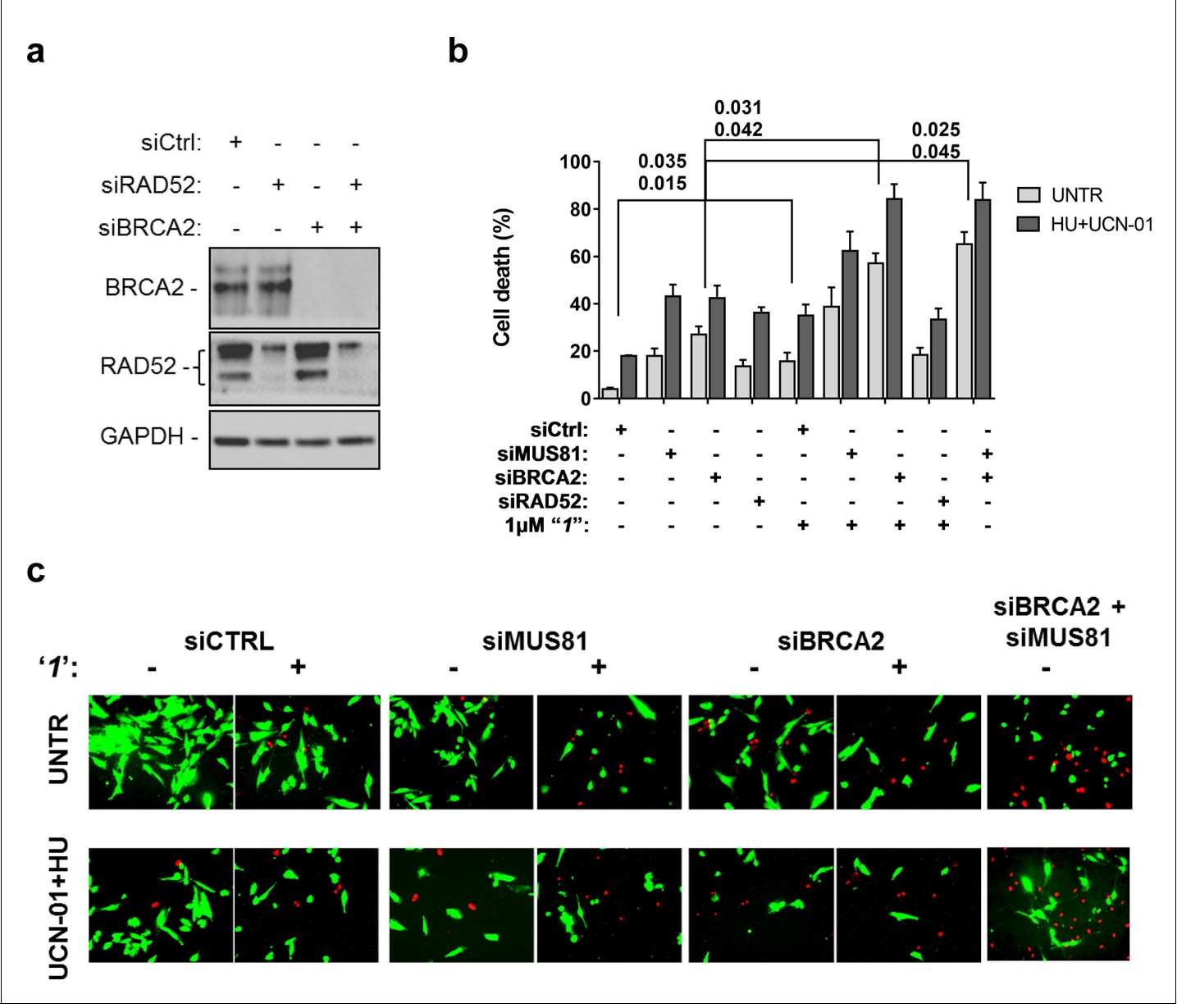

**Figure 6.** Inhibition of ssDNA binding by RAD52 is sufficient to stimulate cell death in the absence of the MUS81 nuclease or BRCA2 tumor suppressor. (**a**) The WB shows the analysis of RNAi. (**b**) Evaluation of cell death after replication stress. Forty eight hours after transfection with the BRCA2 or MUS81 siRNAs, alone or in combination, the GM01604 cells were treated with compound '*1*' or solvent (DMSO). Where indicated, the CHK1 inhibitor UCN-01 and HU was added and the cells were treated for 6h, followed by 18 hr of recovery in drug-free medium. Compound '*1*' was present during the 6h of treatment. Cell viability was evaluated by LIVE/DEAD assay as described in 'Materials and methods'. Data are presented as percentage of dead cells and are mean values from three independent experiments. Error bars represent SEM. The numbers shown in the graph represent the p value; the first p value of each pair refers to untreated cells while the second to the treated cells (2 way ANOVA). (**c**) Representative images of live cells (green) and dead cells (red).

loss of BRCA2 resulting in ~80% of cell death after replication stress (**Figure 6b and c**). Concomitant depletion of BRCA2 and MUS81 also resulted in an additive effect on cell death, even under an unperturbed cell growth (**Figure 6b and c**). To further investigate the effect of '*1*' on cell viability under the conditions of pathological replication, we depleted cells with BRCA2 or RAD52 siRNAs and challenged them with HU for 18h in the presence or absence of '*1*'. As reported in **Figure 7**, and in agreement with previous reports, co-depletion of BRCA2 and RAD52 increased cell death

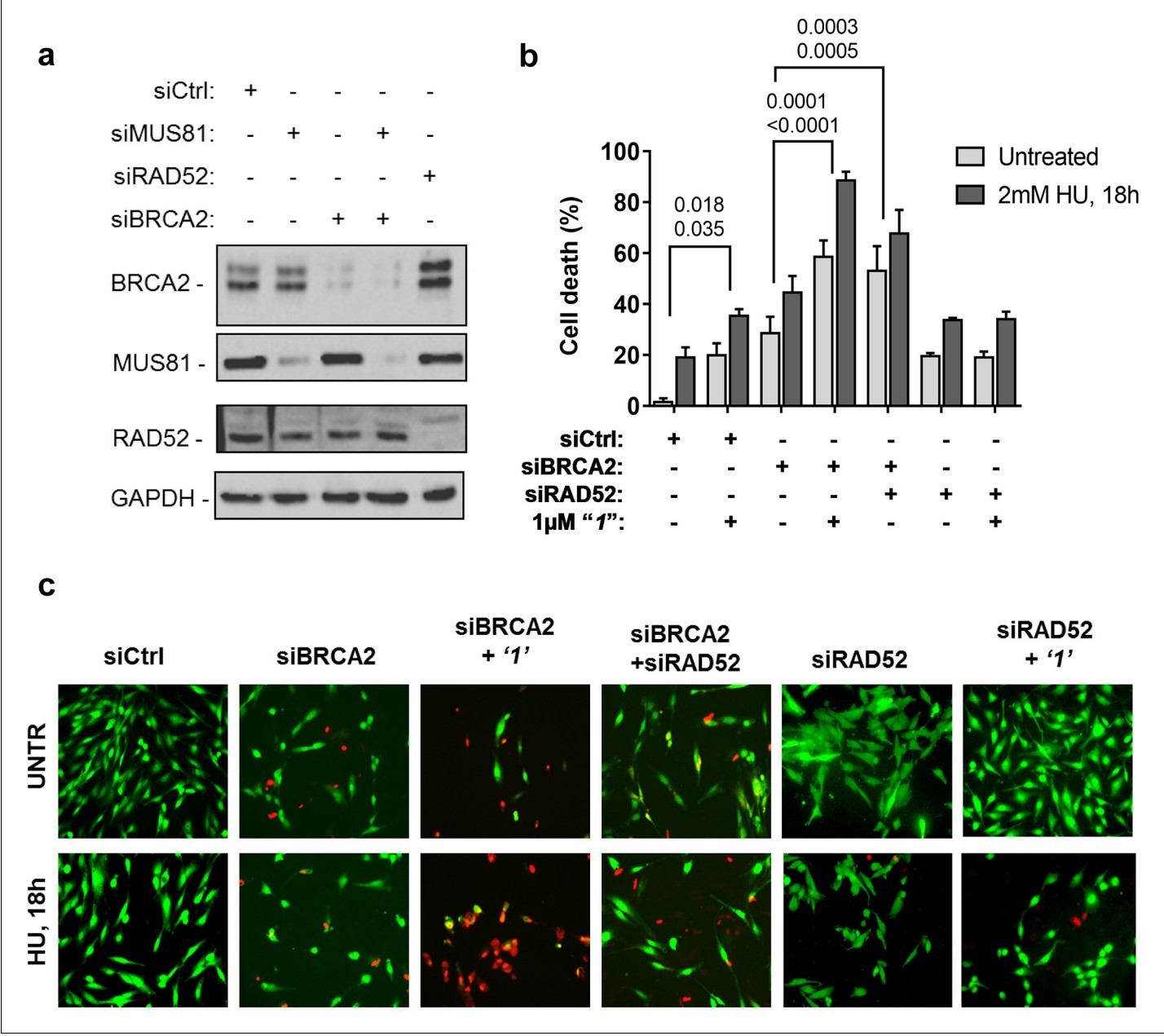

**Figure 7.** Inhibition of ssDNA binding to RAD52 is sufficient to stimulate cell death in the absence of the BRCA2 tumor suppressor. (a) Western blot analysis of BRCA2, RAD52, and GAPDH (loading control) protein levels in GM01604 cells treated with Ctrl, BRCA2, and RAD52 siRNAs. (b) Evaluation of cell death after replication stress in cells treated with '1'. GM01604 cells were transfected with the RAD52 or BRCA2 siRNAs, alone or in combination, and 48h thereafter treated as indicated. The '1' inhibitor or solvent (DMSO) was added to media 1h prior to replication stress. Cell viability was evaluated by LIVE/DEAD assay as described in the 'Materials and methods'. Data are presented as percentage of dead cells and are mean values from three independent experiments. Error bars represent standard error. The numbers shown in the graph represent the p value; the first p value of each pair refers to untreated cells while the second to the treated cells (2 way ANOVA). (c) Representative images: live cells are stained green, while dead cells are red.

with respect to each single depletion. Notably, RAD52 inhibition mimicked RNAi-mediated gene knockdown and induced a substantial increase in cell death of BRCA2-depleted cells after prolonged replication arrest, further supporting the possibility that inhibiting the RAD52-ssDNA interaction may be a useful strategy for targeting BRCA2-deficient tumors.

## In silico screening and discovery of NP-00425: translating structure-activity relationships from HTS into novel inhibitors

We hypothesized that the computationally determined RAD52-'**1**' and RAD52-'**6**' complexes could be used to validate an in silico workflow directed towards identifying a novel inhibitor of the RAD52-ssDNA interaction. This approach should facilitate further discovery of novel drug lead *compounds* that possess similar or improved activities as '**1**' and '**6**', but with a fundamentally different chemical space. Natural products have an unrivaled history in drug discovery, and often represent the first and most significant hits against a metabolic pathway. The AnalytiCon Discovery MEGx Natural Products Screen Library, which is the in silico version of an actual library of purified natural products from plant, fungal and microbial sources, was subjected to an in silico screening campaign (*Figure 8a*). The campaign was designed based on the ability to optimally minimize false positives and false negatives, and to maximize true positives and true negatives. More specifically, the experimental hits identified in the HTS campaign described above, constitute the true positives, while specifically selected decoy compounds constitute the true negatives. The details of this optimization approach, known as Receiver Operator Characteristic (ROC), are described in the Materials and methods section. The ROC procedure in this in silico screening study was designed to challenge the value of the docking and scoring methods by employing decoy compounds (so called 'DUDS' compounds; see Materials and methods section), which possess similar chemical properties, but different topologies than true positives. Importantly, the ROC approach uses the experimental HTS hits to optimize the in silico screening assay *vis-à-vis* minimizing false positives (which are rampant in all docking-based in silico screening approaches). The ROC curve for '**1**' (*Figure 8b*) shows that the optimized in silico selection process is nearly ideal in separating false positives from true positives. Finally, an in silico screen of the AnalytiCon Discovery MEGx Natural Products data base resulted in 9 compounds that had poses with scores better than those of compound '**1**'. The best scoring of those structures was ordered from AnalytiCon Discovery GmbH (Potsdam, Germany) for in vitro inhibition studies. The compound that was identified, **NP-004255** is known as *corilagin*, and is a member of the class of secondary plant metabolites called *ellagitannins*. Corilagin is a macrocyclic ester consisting of three trihydroxylated phenolic moieties. **NP-004255** binds to RAD52 in a similar manner as '**1**' and '**6**', in that it uses a buried interstitial water network, and is able to adopt a conformation that fits nicely into the ssDNA binding groove (*Figure 8d and e*). The binding and inhibitor activity of this prediction was validated by both NMR and FRET-based assays, as described below.

## Natural product NP-00425 physically interacts with RAD52 and RPA proteins, inhibits the RAD52 binding to ssDNA and the ssDNA-RPA complex, but does not affect RAD52-dsDNA interaction or the DNA binding by RPA

The biochemical assays that were carried out for inhibitors '**1**' and '**6**', were repeated for NP-004255 (*Figure 9*). The WaterLOGSY spectra suggest that similar to '**1**' and '**6**', NP-004255 physically interacts with RAD52 (*Figure 9a*), while the FRET-based competition assay (*Figure 9b*) confirmed that this natural product does indeed inhibit the RAD52-ssDNA interaction with an $IC_{50} = 1.5 \pm 0.2$ µM, which is a potency similar to '**1**'. Also similar to '**1**', the macrocycle compound was specific for the RAD52-ssDNA complex and had no effect on the RAD52-dsDNA interaction (*Figure 9b*).

Similar to '**6**', NP-004255 also bound RPA (*Figure 9c*), but did not affect the RPA-ssDNA complex (*Figure 9d*). It did, however, inhibit RAD52 binding to RPA-coated ssDNA with an $IC_{50} = 0.5 \pm 0.1$ µM, *i.e.* it was more effective in perturbing this interaction that that involving the protein-free ssDNA.

## Discussion

Maintenance of genetic integrity, as well as the ability to accurately and timely repair damaged DNA and complete DNA replication are essential for all living organisms (*Heyer, 2015*; *Abbas et al., 2013*). While these basic processes and the central protein players are conserved, significant variation exists between eukaryotic lineages. The mechanisms that ensure faithful DNA replication and repair are exceedingly more complex in mammalian cells compared to simpler eukaryotes with more

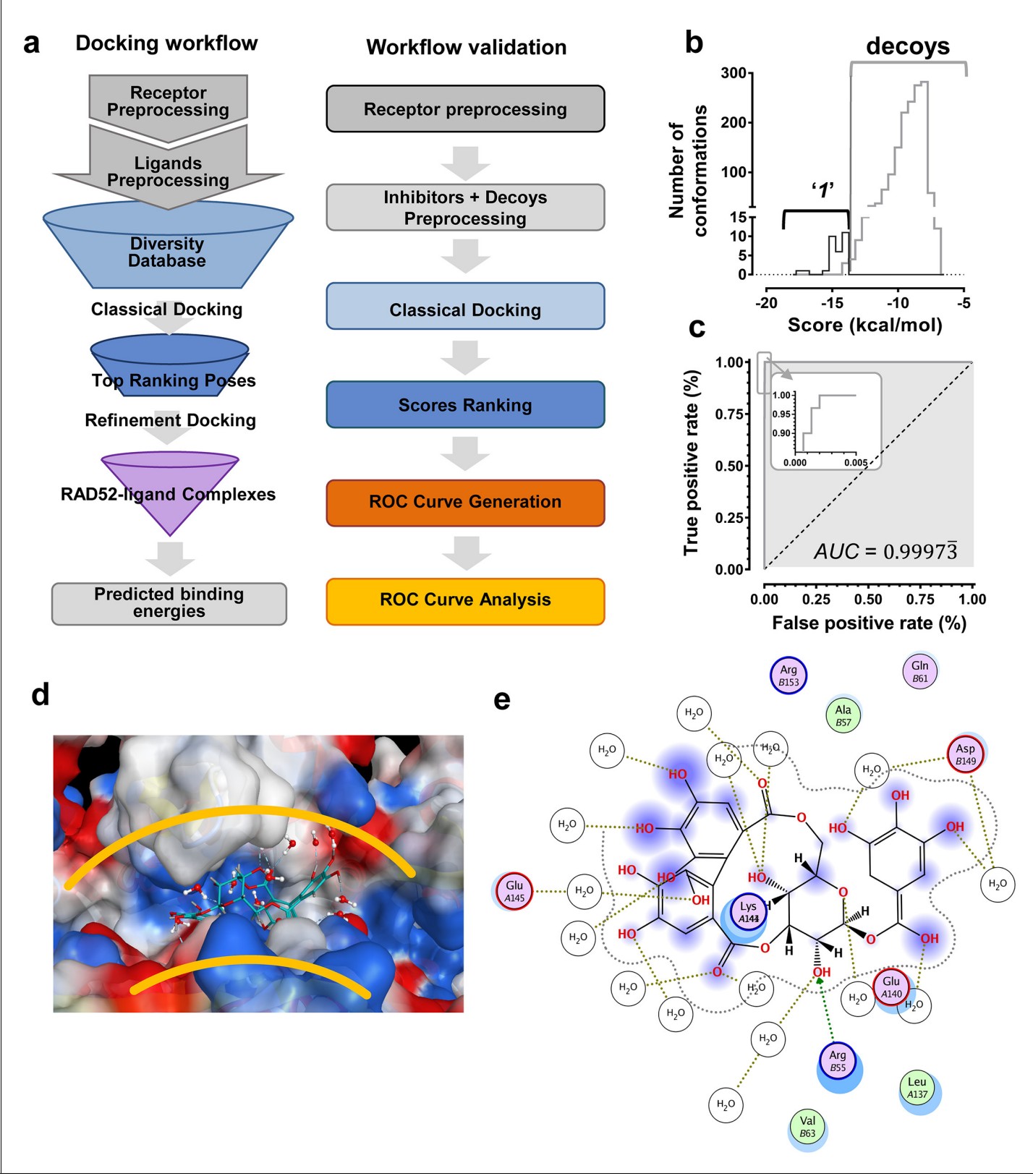

**Figure 8.** In silico screening campaign identifies novel small molecule that inhibits the RAD52-ssDNA interaction. (**a**) Docking workflow involved RAD52-NTD undecameric ring (PDB 1KNO) pre-processing, AnalytiCon Discovery MEGx Natural Products Screening Library pre-processing, and classical docking using the Dock utility of MOE. Top ranking poses (those with the lowest energy scores from the London dG scoring function) were subjected

*Figure 8 continued on next page*

*Figure 8 continued*

to a refining docking step involving force field-based energy minimization. From these complexes, predicted binding free energies were calculated. Workflow validation involved RAD52-NTD pre-processing, pre-processing of '*1*' and associated decoys (DUD-E), followed by classical docking and score ranking. ROC curves were then generated and analyzed. Scores of the conformations of inhibitor compounds and of conformations of their respective decoy compounds were compared. (**b**) Docking scores (kcal/mol) for the individual conformations of compound '*1*' and decoy compounds were binned and plotted as histograms. The low docking scores, as indicated by the more negative predicted free energies of '*1*' when compared to decoys, indicate more favorable poses, and highlight a distinct separation between true positives and true negatives. (**c**) Receiving-operating characteristic (ROC) curve shows that the classifier used, *i.e.* the scoring function, was close to optimal in distinguishing compound '*1*' conformations from those of decoys confirmed by AUC analysis yielding a value of 0.9973. (**d**) Electrostatic surface potential of three monomers of the RAD52-NTD undecameric ring (PDB 1KNO) depicting **NP-004255** within the ssDNA binding groove. (**e**). MOE ligand maps highlight water mediated interactions with E145 and D149 as well as via hydrogen bonding with amino acids R55.

alternative interconnected pathways that may share proteins as well as regulatory enzymes. HR and the pathways that employ the machinery of HR are expected to be responsible for the most accurate repair of the most deleterious DNA lesions including DSBs, DNA interstrand cross-links, and damaged replication forks (*Head, 2010*; *Li and Heyer, 2008*; *Couedel et al., 2004*; *Moynahan and Jasin, 2010*; *Jasin and Rothstein, 2013*).

In yeast, Rad52 functions as a recombination mediator by facilitating replacement of RPA with Rad51 recombinase on ssDNA and thereby allowing the formation of the Rad51 nucleoprotein filament, which is the active species in the DNA strand exchange reaction. Analogously, RAD51 nucleoprotein filament formation and its activity in human cells is facilitated by a recombination mediator, BRCA2 (*Xia et al., 2001*; *Yang et al., 2005*; *Carreira et al., 2009*) with multiple RAD51 paralogs playing roles in ensuring assembly and stability of the active RAD51 nucleoprotein filament (*Yang et al., 2005*; *Chun et al., 2013*). Whether, and how, human RAD52 substitutes for BRCA2 mediator activity remains unclear. Synthetic lethality between BRCA2 defects and RAD52 depletion suggests that either RAD52 is indeed a recombination mediator, or that it participates in an alternative pathway that becomes prominent in the absence of BRCA2 function, such as for example SSA (single-strand annealing) (*Feng et al., 2011*; *Lok and Powell, 2012*). More intriguingly, RAD52 depletion is also synthetically lethal with defects in BRCA1, a tumor suppressor that acts upstream of BRCA2 in HR and at the branch point in the DSB repair that promotes homology-directed DNA repair through HR or SSA over the NHEJ (non-homologous end joining) (*Singleton et al., 2002*). It is unknown which pathway(s) allow survival and proliferation of BRCA-deficient cells. These pathways, however, have to depend on the activities or interactions of RAD52. We showed recently that human RAD52 plays an important role in allowing cellular recovery under conditions of pathological replication (*Murfuni et al., 2013*). Similarly, a sub-pathway of HR that is Rad51 (Rhp51) independent, but Mus81/Eme1/Rad52 (Rad22) dependent has been described in yeast and represents an important mechanism of DNA repair during replication in fission yeast (*Doe et al., 2004*; *Vejrup-Hansen et al., 2011*). Whether this pathway, at least in part, compensates for the BRCA-deficiency in human cells remains to be determined.

We chose to target the well characterized ssDNA binding activity of RAD52, which we expected to underlie RAD52 functions both in supporting replication and in promoting the survival of BRCA-deficient cells. The ssDNA-binding groove of RAD52 is an interesting target for small-molecule binding in that it spans the circumference of the RAD52 oligomeric ring and offers a repetitive pattern of potential binding pockets. This deep and circular groove surprisingly yields reasonable druggability scores, as described in the Results section. Nevertheless, it is a highly exotic cavity, and very distinct from enzyme and receptor pockets. The ssDNA binding groove consists of an alternating arrangement of hydrophobic and hydrophilic regions. Twelve compounds that inhibit RAD52-ssDNA interaction identified in this study (*Table 1*) are predicted to bind within the ssDNA binding groove of RAD52 ring. In retrospect, it is not surprising that molecules such as the current suite of polyphenols have high affinity for this cavity. Additionally, although there are a number of hydrophobic regions in the ssDNA binding groove, one does not see the kind of significant desolvation that is usually found in enzyme active sites. Nevertheless, a number of waters are revealed in the crystal structure, and are maintained in the docking simulations. However, the nature and importance of these water networks to ligand optimization is not known. It appears there is significant room for improvement in

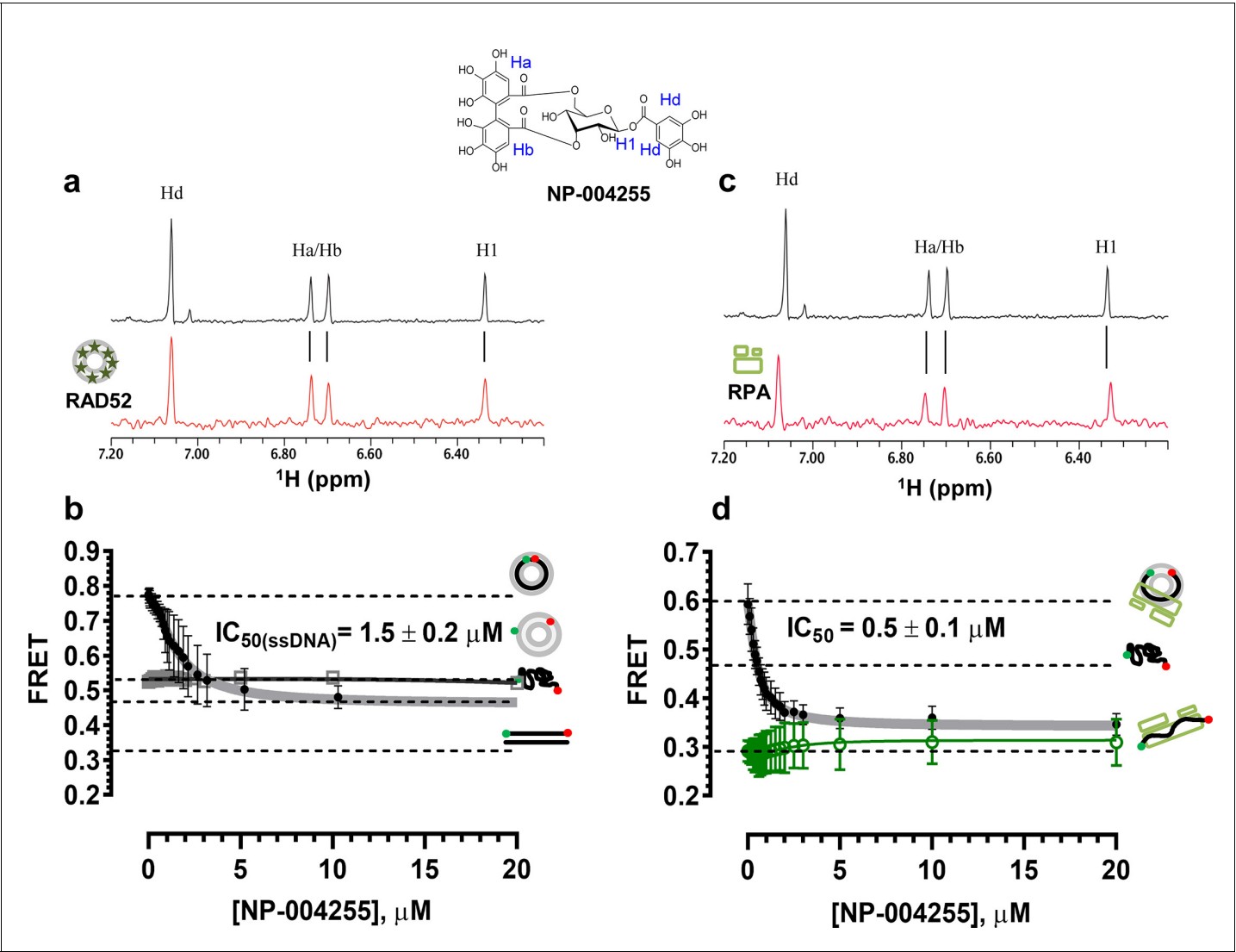

**Figure 9.** Biochemical characterization of NP-004255. (**a**) Aromatic region of the 1D $^1$H NMR spectrum of compound **NP-004255** alone (black) and the WaterLOGSY spectrum of 40 μM compound **NP-004255** in the presence of 3.3 μM RAD52 (red). The nonexchangeable proton peaks are labeled using atom names as indicated on the structure of compound **NP-004255**. (**b**) IC$_{50}$ values for inhibition of ssDNA binding and wrapping were determined using FRET-based assays that follow the change in geometry of a Cy3-dT30-Cy5 substrate (black circles). The computed IC$_{50}$ value is shown above the curve. Titration of the RAD52-dsDNA with **NP-004255** (grey boxes) shows that this inhibitor does not perturb the RAD52-dsDNA interaction. (**c**) Aromatic region of the 1D $^1$H NMR spectrum of compound **NP-004255** alone (black) and the WaterLOGSY spectrum of 40 μM compound **NP-004255** in the presence of 3.3 μM RPA (red). (**d**) Titration of the RAD52-RPA- Cy3-dT$_{30}$-Cy5 complex with **NP-004255** (black circles). The computed IC$_{50}$ value is shown below the curve. Green squares show titration of the RPA- Cy3-dT$_{30}$-Cy5 complex with **NP-004255** indicating **NP-004255** does not perturb the RPA-ssDNA interaction.

terms of matching the shape of the binding groove with the van der Waals surface of prospective ligands. It will be interesting to see whether such chemical space is extant or may be designed to optimize this unusual surface. The computational studies on the complexation of '*1*' and '*6*' with RAD52 indicate the presence of a ubiquitous layer of interstitial water interactions with RAD52, yet these ligands are almost completely shielded from bulk solvent. The presence of these extensive interstitial water contacts further complicates hypotheses concerning which, if any, RAD52 functional groups are dominating the binding energy contacts. Rather, it may be that our HTS-generated hits possess the right combination of Van der Waals shape complementarity, and the ability to be both hydrogen bond donors and acceptors (with both interstitial waters and functional moieties) in the

narrow DNA binding groove. Indeed, it may be that this shape complementarity and the ability to utilize the resident waters dominates the binding determinants (both for the identified ligands, as well as the native ssDNA substrate). Our iterative approach of compound discovery followed by the in silico screening was clearly successful in expanding the chemical space of our lead compounds, but more importantly provides a platform for strengthening the structure-activity relationship in an exceedingly challenging target pocket. Indeed, the discovery of the secondary plant metabolite, **NP-004255**, a macrocycle, as a means to effectively compete with a native substrate macromolecule (ssDNA and the ssDNA-RPA complex) may prove to be a new strategy in the field of disrupting protein-nucleic acid interactions. It has to be noted, however, that although our work provides a solid understanding of how shape complementarity and utilization of interstitial water networks drives complexation, there are limitations to these approaches. Specifically, the current experimental and computational studies do not address the underlying thermodynamics and kinetics that control ligand competition in the ssDNA binding groove. A fascinating aspect of the competition between small molecules and macromolecules for an extended and partially solvated binding pocket, as in RAD52, is the extent to which small molecules may exploit the potential enthalpy-entropy compensation of a large and floppy native DNA ligand. Additionally, residence times of lead compounds may be a critical factor in the successful design of therapeutic lead compounds. These factors will be addressed in the future studies that will be focused on the thermodynamics and kinetics of small molecule binding to the RAD52 protein, and in the studies that will confirm the ligand placement through high resolution structures.

Recently, Chandramouly and colleagues (*Chandramouly et al., 2015*) identified a small-molecule RAD52 inhibitor, 6-hydroxy-DL-dopa, that acts differently from the molecules reported here. This inhibitor interferes with RAD52 oligomerization and the supramolecular assembly by an unresolved mechanism. It may act by binding at the RAD52 monomer-monomer interface, or at a different site on the protein and act allosterically. The existence of the distinct classes of RAD52 inhibitors, exemplified by '1' and 6-hydroxy-DL-dopa, suggests that disrupting the RAD52-ssDNA interaction or the integrity of the RAD52 oligomeric ring bears negative consequences for the RAD52 cellular functions. Considering that the efficient homology search and the DNA strand annealing requires the two complementary DNA strands (or the complementary ssDNA-RPA complexes) to be wrapped around the two different RAD52 oligomeric rings (*Grimme et al., 2010*; *Rothenberg et al., 2008*), this is not surprising, and offers an exciting opportunities for development of more specific and potent agents for targeting recombination-deficient tumors.

While this manuscript was under review, two studies reporting small-molecule inhibitors of RAD52 were published. In the first study, Huang *et al* (*Huang et al., 2016*) carried out an HTS campaign to identify 17 compounds that inhibit RAD52-mediated annealing in vitro with $IC_{50}$ values ranging between 1.7 and 17 µM, physically bind RAD52, and selectively, albeit at high concentrations, inhibit the single-strand annealing pathway of DSB repair over homologous recombination. In another study, Sullivan *et al* (*Sullivan et al., 2016*) reported an in silico docking screen of a library of drug-like compounds. Among 36 predicted small-molecules, 9 compounds inhibited RAD52-ssDNA interaction in vitro, and 1 in cells. As with all previous publications, the authors screened different libraries, resulting in compounds that represent a different chemical space from those that emerged from our campaign.

Since the expected role of RAD52 in the recovery of stalled replication forks in the absence of cellular checkpoint is to produce an intermediate that can be cleaved by MUS81/EME1 nuclease, we predicted that the ssDNA binding/annealing activity of RAD52 is required to fulfill this role. As expected, we found that '1' and '6' recapitulate inhibition of DSB formation by siRNA mediated RAD52 depletion (*Figure 5*). In the case of '1', 1 µM of the inhibitor was sufficient to achieve the same level in DSB reduction as siRNA treatment. Notably, no further inhibition was observed when the cells were treated with both siRNA and the small-molecule inhibitor suggesting that the effect is specific to RAD52. A higher concentration of '6' was required to achieve the level of reduction in the RAD52/MUS81 dependent DSBs comparable with siRNA depletion of RAD52. This may be due to metabolic instability of this compound or due to potential off-target binding. For this reason, we placed an increased focus on '1'. The ability of '1' to inhibit DSB formation, which is required for the recovery of damaged replication forks in checkpoint deficient cells confirms that the ability of RAD52 to bind ssDNA is required for MUS81-dependent cleavage at stalled replication forks. Moreover, it is consistent with the mechanism we previously proposed whereby, in these cells, RAD52

used its ssDNA-binding activity to create a substrate for MUS81/EME1 and to recruit this structure selective nuclease. Consistent with our previous finding (*Murfuni et al., 2013*), RAD52 inhibition with '*1*' acted additively with MUS81 depletion eliciting an effect comparable with RAD52 depletion. At 1 µM concentration of the inhibitor, approximately 40% of untreated and 60% of checkpoint-deficient, HU treated cells were dead (*Figure 6b*). This is notable because the inhibitor interferes only with the biochemical function of RAD52, namely its ability to bind ssDNA, while leaving the protein itself and its cellular concentration unperturbed, and also because even temporary loss of this biochemical activity during the exposure to replication stress is sufficient to exert the additive effect on viability. This result also illustrates the potentially powerful utility of these inhibitors in elucidating the function of RAD52 in the cell. We observed that inhibition of RAD52 during replication stress, which is induced by blocking DNA synthesis in the absence of the CHK1 activity, in a MUS81 knockdown background results in a comparable effect on viability as the concomitant depletion of both proteins. This observation strengthens the hypothesis that loss of MUS81 and RAD52 produces an additive lethal effect on replication stress (*Murfuni et al., 2013*) because, while RAD52 and MUS81 collaborate to resolve demised forks, MUS81 is subsequently required for resolution of recombination intermediates in a RAD52-independent pathway (*Figure 5—figure supplement 1*). More interestingly, '*1*' was able to act at least additively with BRCA2 knock-down (*Figure 6b*). An increase in cell death when BRCA2 depletion was combined with '*1*' was comparable to or even exceeded that of MUS81 depletion by siRNA. Notably, the effect of '*1*' was further enhanced by replication stress induced by short HU treatment and concomitant CHK1 inhibition, as well as by a prolonged exposure to HU. Treatments inducing replication stress are widely used in cancer therapy (e.g. CPT, Gemcitabin). Therefore, RAD52 inhibitors could be useful in combination with drugs which elicit replication stress. Tumors in which MUS81 is mutated or downregulated have been described (*Wu et al., 2011*). While it is unclear whether the role RAD52 plays in supporting the survival of the MUS81-deficient cells is akin to its role in supporting viability in the absence of BRCA1, BRCA2 or PALB2, RAD52 inhibitors may represent a new means of treatment for the MUS81-deficient tumors as well as the BRCA-deficient tumors. In addition to its obvious uses in cancer therapy, the RAD52-ssDNA binding inhibitors can be utilized as molecular probes to assist in distinguishing the cellular pathways that depend on the main biochemical activity of RAD52. RAD52 may act together with other HR proteins, such as RAD51 paralogs to ensure formation of the active RAD51 nucleoprotein filament during RAD51-dependent HR. An understanding of the common players which might bind RAD52 in the absence of BRCA1 or BRCA2 and how they are regulated in the BRCA-deficient cells may require development of specific inhibitors of RAD52 protein-protein interactions and/or combining our inhibitors with other treatments that challenge homology directed DNA repair and replication.

## Materials and methods

### Materials

The HPLC purified ssDNA substrate (Cy3-dT$_{30}$-Cy5), Target28Cy3 (T-28) (5'-ATAGTTATGGTGAG-GACCC/iCy3/CTTTGTTTC-3'), Probe28Cy5 (P-28) (5' GAAACAAAGGGGTCC/iCy5/ TCACCATAAC-TAT-3') Oligo28-REV (5'-(**Cy5**)-GCAATTAAGCTCAAGCCATCCGCAACG-(**Cy3**)-3', Cy3-Oligo28-Cy5 (5'-CGTTGCGGATGGCTTAGAGCTTAATTGC-3', and Poly dT100 were purchased from Integrated DNA Technologies (Coralville, IA, USA). All chemicals were reagent grade (Sigma-Aldrich, St. Louis, MO). All compounds were purchased from MicroSource Discovery systems, Inc (Gaylordsville, CT, USA) and Sigma-Aldrich. The NP-004255 was from AnalytiCon Discovery GmbH (Potsdam, Germany). Purity and structures of the purchased compounds were assessed from 1H NMR spectra collected on a Varian Unity Inova 600 MHz NMR spectrometer at 0.5 mM concentrations diluted into DMSO-d6.

### Proteins

The 6xHis-tagged human RAD52 protein was expressed and purified as previously described (*Rothenberg et al., 2008*), except a size exclusion chromatography (HiPrep 16/60 Sephacryl S-300 HR GE Healthcare Life Sciences, Pittsburgh, PA, USA) step was added between the heparin and Resource S columns to remove low molecular weight impurities. RAD52 protein concentration was

determined by measuring absorbance at 280 nm using extinction coefficient 40,380 $M^{-1}$ $cm^{-1}$. RPA protein was purified as described in (*Henricksen et al., 1994*; *Grimme et al., 2010*) (and its concentration was determined by measuring absorbance at 280 nm using extinction coefficient 84,000 $M^{-1}$ $cm^{-1}$.

## High-throughput screening assay for RAD52-ssDNA binding inhibition

HTS against the MicroSource Spectrum collection (Microsource, Gaylordsville, CT) was performed in nine 384 well plates. All measurements were carried out in the RAD52-HTS buffer containing 20 mM Hepes pH7.5, 1 mM DTT, and 0.1 mg/mL BSA. Each 384 well plate contained two columns of negative and positive controls as follow: Columns 1 and 24 were the positive controls, which contained 100 nM RAD52 (monomers) and 15 nM (molecules) Cy3-$dT_{30}$-Cy5 ssDNA in the RAD52-HTS buffer. Columns 2 and 23 in addition to 100 nM RAD52 and 15 nM Cy3-$dT_{30}$-Cy5 also contained 10 nM poly(dT)-100. These were designated as negative controls as 10 nM poly(dT)-100 was sufficient to fully inhibit formation of the wrapped RAD52- Cy3-$dT_{30}$-Cy5 complex under the selected experimental conditions (data not shown). Using a Multiflo dispenser (Biotek), 50 µL of the positive and negative controls were dispensed into their respective wells. Then 15 nM Cy3-$dT_{30}$-Cy5 and 100 nM RAD52 in RAD52-HTS buffer were dispensed into each well. Next, 1 µL of each compound dissolved in DMSO at 833 µM concentration (for a final concentration of 15 µM compound) was dispensed into the wells in columns 3 – 22 using a Microlab Star liquid handling robot (Hamilton) and mixed 3 times. Thus, 320 compounds were assayed per 384 well. The plate was incubated at 25°C for 30 min and the fluorescent signal of the Cy3 ($\lambda_{ex(Cy3)}$= 530 nm; $\lambda_{em(Cy3)}$= 565 nm) and the Cy5 ($\lambda_{em(Cy5)}$= 660 nm) dyes were recorded using a Wallac, Envision Manager. The apparent FRET was calculated as

$$FRET_{app} = \frac{I_{Cy5}}{I_{Cy5} + I_{Cy3}}$$

Assay performance was assessed across the screen using the following parameters: The signal-to-noise ratio $S/N = \frac{(\mu_n - \mu_p)}{SD_n}$, the signal-to-background ratio $S/B = \frac{\mu_n}{\mu_p}$, and the Z'-factor $Z' = 1 - \frac{3*(SD_n + SD_p)}{(\mu_n - \mu_p)}$, where $SD_p$ and $SD_n$ are standard deviations, and $\mu_n$ and $\mu_p$ are means of the negative and positive control (*Zhang et al., 1999*).

Compounds from the wells that showed apparent FRET values at least 5 SD lower than the positive control were considered potential hits and were selected for the follow up analysis. Ninety six compounds were re-screened to assess reproducibility of hits. Twenty two of these compounds were removed due to high background signal. Twelve compounds, which showed a reproducible reduction in FRET from a screening of the individual plates and re-screening in cherry picked plates, were selected for biochemical validation.

## WaterLOGSY NMR analysis of the compound binding to RAD52 and RPA proteins

Compound binding to RAD52 and RPA proteins was analyzed using water-ligand observation with gradient spectroscopy (WaterLOGSY) NMR experiments (*Dalvit et al., 2001*, *2000*). The Water-LOGSY spectra of compounds in the presence of RAD52 or RPA were acquired using a water NOE mixing time of 1 s and a $T_2$ relaxation filter of 50 ms just before data acquisition to suppress the broad signals derived from protein. The protein buffer used in the experiments contains 10 mM Tris-d11, 75 mM KCl, 0.25 mM EDTA, pH 7.5, and 10% $D_2O$. All NMR data were acquired on a Bruker Avance II 800 MHz NMR spectrometer equipped with a sensitive cryoprobe and recorded at 25°C. The $^1H$ chemical shifts were referenced to 2,2-dimethyl-2-silapentane-5-sulfonate (DSS). NMR spectra were processed using NMR Pipe (*Delaglio et al., 1995*) and analysed using NMR View (*Johnson and Blevins, 1994*).

## FRET-based DNA binding and annealing assays

FRET-based ssDNA binding, dsDNA binding, and annealing assays were carried out as previously described (*Grimme et al., 2010*; *Grimme and Spies, 2011*) using Cary Eclipse spectrofluorimeter (Varian) at 25°C in buffer containing 30 mM Tris-Acetate pH7.5, 1 mM DTT, and 0.1 mg/mL BSA. Cy3 dye was excited at 530 nm and its emission was monitored at 565 nm. Emission of Cy5 acceptor

fluorophore excited through the energy transfer from Cy3 donor is monitored at 660 nm simultaneously with emission of Cy3 dye. Both the excitation and the emission slit widths were set to 10 nm.

To confirm that selected compounds inhibit RAD52 mediated binding and wrapping of ssDNA, compounds were titrated into stoichiometric complex containing 1 nM T30 and 8 nM RAD52. All experiments were performed in triplicates, and the data are shown as averages and standard deviations for three independent measurements. To remove possible experimental artifacts associated with chromogenic or fluorogenic compounds, as well as with the compounds that may quench or enhance Cy3 or Cy5 fluorescence, we also performed control titrations whereby we titrated each compound into 1 nM Cy3-dT$_{30}$-Cy5 in the absence of RAD52. For each compound concentration we subtracted the difference in the FRET signal in the presence and absence of the compound from the respective FRET signal in the presence of RAD52. The FRET signal corrected for the compound fluorescence was calculated using the equation:

$$FRET_{app} = \frac{4.2 * I_{Cy5}}{4.2 * I_{Cy5} + 1.7 * I_{Cy3}}$$

as previously described (*Grimme et al., 2010*; *Grimme and Spies, 2011*) and plotted as a function of compound concentration and fitted to the following inhibition model"

$$FRET([small\ molecule]) = \frac{FRET_0 - FRET_{min}}{1 + 10^{(LogIC50 - Log([small\ molecule]) * HillSlope)}},$$

where FRET$_0$ is the initial FRET value of the RAD52-ssDNA complex in the absence of the compound and FRET$_{min}$ is the FRET value at complete inhibition. FRET values were calculated as an average of three or more independent annealing reactions plotted against the concentration of the compound. Inhibition of the RAD52-dsDNA interaction was assayed in a similar experiments, except the stoichiometric complexes containing 1 nM molecules of Cy3-Oligo28-Cy5 duplex DNA and 10 nM RAD52.

To assess if selected compounds inhibit RPA-ssDNA mediated binding and wrapping by RAD52 compounds were titrated into stoichiometric complexes containing 1 nM T30, 1 nM RPA, and 10 nM RAD52. In all experiments, the FRET values for each data point were corrected for the effects of the compounds on the respective substrate in the absence of RAD52.

Annealing of complementary oligonucleotides by RAD52 was monitored under identical conditions as the binding assays described above. For each assay, the reaction master mixture containing 8 nM RAD52 protein in the presence and absence of the compounds at varying concentrations was prepared at room temperature and divided into two half reactions. Following baseline buffer and protein measurements, 0.5 nM of T-28 ssDNA substrate was added to the reaction cuvette and the signal was allowed to stabilize. The annealing reaction was initiated upon addition of the second half-reaction pre-incubated with 0.5 nM P-28 ssDNA substrate. The fluorescence of Cy3 and Cy5 were measured simultaneously over the reaction time course (500 s). FRET$_{app}$ values were calculated as an average of three or more independent annealing reactions plotted against time (s). The average FRET values were fitted to a double exponential to calculate the final extent of annealing using Graphpad Prism4 software. The calculated annealing extent was plotted as a function of compound concentration and fitted to the same model as we used to determine IC$_{50}$ values for the inhibition of DNA annealing.

## Docking, molecular mechanics (MM) and generalized born (GB)/surface area (SA) (MMGB/SA)-based free energy scoring for RAD52 ligands

Our initial computational workflow employed a combination of classical docking, using the Triangle Matcher approach and scoring using the London dG scoring function (an empirical scoring function which attempts approximate the binding energy of the docked ligand) in MOE 2013.08 (*Chemical Computing Group, 2013*), followed by force field (MMFF94x [*Halgren, 1996*])-based ligand refinement and finally rescoring using an MM/GBSA-based approach (which is described in more detail below). Initially, top scoring poses in either Triangle Matcher (Placement), London dG (affinity scoring function) or MM/GBSA (physics-based scoring) were retained for further analysis. Often the top scoring poses from each metric were highly distinct from one another, suggesting that a generally poor consensus between the different metrics used in this early phase of the work

flow. This lack of consensus in the scoring of the possible ligand binding in the sub-pockets within the DNA-binding groove of RAD52 (PDB: 1KNO) motivated us to apply the much more computationally rigorous all atom-simulated annealing studies, which are detailed below. All lead ligands were subjected to docking using MOE 2013.08 to a portion of the DNA binding groove of RAD52 spanning nearly a quarter of the circumference (3 adjacent monomers of the protein ring). The top 30 poses for each docked and scored (London dG scoring function) were subjected to energy minimization with a rigid RAD52 receptor using the MMFF94x force field, followed by rescoring (in order to estimate the ΔG of binding) of each distinct pose with the MM/GBSA methodology (*Naïm et al., 2007*), which includes an implicit solvation energy calculation and captures changes in the solvent exposed surface area of the pose, which is a highly parameterized version of the popular MM/PBSA and MM/GBSA methodologies (*Steinbrecher and Labahn, 2010*; *Wang and Kollman, 2000*).

### All atom-simulated annealing energy minimization with the YASARA2 knowledge-based forcefield, and rescoring with VINA

The all atom-simulated annealing energy minimization (here referred to SAEM for brevity), which is followed by a local docking protocol (as described below) is a customized protocol that was automated with a script using the Python-based Yanaconda scripting language, and use of the Yamber03 knowledge-based force field (*Krieger et al., 2004*). Briefly, each complex was placed into a simulation cell and solvated, and charge-neutralized to yield physiological conditions, followed by an optimization of the solvent and H-bonding network, and finally a phased simulated annealing minimization was performed (a similar process is described in *Whalen et al. (2011)*). No restraints were placed in any of these systems (i.e. all atoms in the ligand and the entire RAD52 complex, ions and solvent were free to move in the simulation). The affinity of the ligand in this optimized complex was then determined by scoring with AutoDock VINA (*Trott and Olson, 2010*). Water molecules that were interstitial were automatically retained in the VINA scoring.

### Neutral comet assay

To induce RAD52-MUS81-dependent cleavage at arrested replication forks (*Murfuni et al., 2013*), hTERT-immortalized wild-type human fibroblasts (GM01604) were treated with 2 mM HU and 300 nM UCN01 for 6h, in the presence and absence of varying concentrations of '1' or '6'. Where indicated, the GM01604 cells were cells were transfected with siRNAs directed against GFP (Ctrl), or against RAD52 (Qiagen) 48 hr prior to induction of replication stress and/or inhibitor treatment. After that cells were subjected to neutral comet assay as described in Murfuni et al (*Murfuni et al., 2013*). Slides were analyzed by a computerized image analysis system (Comet IV, Perceptive UK). To assess the quantity of DNA damage, computer-generated tail moment values (tail length x fraction of total DNA in the tail) were used. Apoptotic cells (smaller comet head and extremely larger comet tail) were excluded from the analysis to avoid artificial enhancement of tail moment. A minimum of 100 cells were analyzed for each compound concentration point.

### Cell viability live/dead assays

GM01604 cells were transfected with siRNAs directed against GFP (Ctrl), or against MUS81 (Qiagen), BRCA2 (Sigma-Aldrich), and RAD52 (Qiagen) 48 hr prior to addition of 1 μM '1'. Where indicated, the conditions of pathological replication were induced by treating cells with 2 mM HU and 300 nM UCN01 for 6 hr or by 18 hr treatment with 2 mM HU, in the presence or absence of '1'. Viability was evaluated by the LIVE/DEAD assay (Sigma-Aldrich) according to the manufacturer's instructions. Cell number was counted in randomly chosen fields and expressed as percent of dead cells (number of red nuclear stained cells divided by the total cell number) corrected for the cell loss observed in the population. For each time point, at least 200 cells were counted.

### In silico screening leading to identification of novel inhibitor NP-004255 using SAR from HTS

The AnalytiCon Discovery MEGx Natural Products Screen Library, which is the in silico version of an actual library of purified natural products from plant, fungal and microbial sources, which is available for purchase, was subjected to an in silico screening campaign. The campaign was designed based on the ability to optimally minimize false positives and false negatives, and to maximize true

positives and true negatives. More specifically, the experimental hits identified in the HTS campaign described above, constitute the true positives, while specifically selected decoy compounds constitute the true negatives. Decoy compounds were generated using the Database of Useful Decoys – Enhanced (DUD–E) website (*Mysinger et al., 2012*). Decoys are compounds that resemble active ligands in physicochemical properties, but are distinct in chemical topology to true binders, so that separation bias is avoided (*Huang et al., 2006*). Decoys are property-matched to compounds of interest using molecular weight, estimated water-octanol partition coefficient (miLogP), rotatable bonds, hydrogen bond acceptors, hydrogen bond donors, and net charge (*Mysinger et al., 2012*). An average of 50 decoys are obtained for each ligand.

In order to validate the selected protocol (*Figure 1b*), we employed a statistical method in which 'Receiver Operating Characteristic' or ROC curves are used to optimize the balance of true positives, false positives, true negatives and false negatives (*Varnek et al., 2008*). ROC curves were created using MatLab (R2015a; Mathworks, Natick, MA, USA) from scoring ranks of active versus inactive poses for each of the best HTS hits (*Varnek et al., 2008*). This plot represents the percentage of true positives versus percentage of false positives for a wide range of choices of score cutoffs. This procedure also allows the determination of the best score threshold for cutoff of compounds regarding the particular protein target.

## Specific docking and scoring procedures

A database containing the AnalytiCon Natural Products compound library, and a control selected from the initial HTS hits, were created and preprocessed for virtual docking (as described above). The top 30 final poses, generated using the Dock utility of MOE (as described above) were written to an output database. Poses of all compounds were ranked based on their scores. Compounds with the poses most favorable to binding, i.e. the poses with the lowest energy scores from the London dG scoring function were selected for further analysis. Those poses with better scores than the highest scoring pose of our control (ie, a 'true positive' in the ROC curve context) were selected, and then subjected to a refining docking step involving force field-based energy minimization with the MMFF94x force field in MOE. Binding energies were ranked, and evaluated.

## Construction of receiver operator characteristic (ROC) curves

The docking scores (kcal/mol) were used for determining the ROC threshold values (see (*DeLong et al., 1988*), for precise description of the how to determine the threshold value). Each original compound of interest and its poses were to be the only 'predicted positives', and the DUDs (decoys) and its poses were to be the 'predicted negatives'; any poses above the threshold were to be the 'actual positives' and the poses below the threshold were to be the 'actual negatives'. The curves were analyzed using the metric of the area under the curves (AUC) (*DeLong et al., 1988*). The scores of the poses for the most active compounds exhibited bimodal frequency distribution (*Figure 8b*), and the docking protocols' ability to distinguish between active compounds and decoys was verified (*Figure 8c*). We have determined a protocol for sorting compounds of interest among a database that provides reliable results with high cutoff limits.

## Acknowledgement

This work was supported by the American Cancer Society Research Scholar Grant (RSG-09-182-01-DMC) to MS, by the NIH R01-GM097373 to MAS, by the University of Iowa College of Medicine pilot HTS grant to MAS and MS, by the Italian Association of Cancer Research (AIRC, IG13398) and Nando-Peretti Foundation (2012-113) grants to PP and by the NIH S10 (1S10RR029274-01) grant supporting the UIHTS facility. LFdaSC is supported by Coordenação de Aperfeiçoamento de Pessoal de Nível Superior (CAPES) Brazil. The funders had no role in study design, data collection and analysis, decision to publish, or preparation of the manuscript. We thank Drs. Todd Washington, Miles Pufall and members of M Spies' lab for critical reading of the manuscript and for valuable discussions.

## Additional information

### Funding

| Funder | Grant reference number | Author |
|---|---|---|
| Coordenação de Aperfeiçoamento de Pessoal de Nível Superior | | Laura Folly da Silva Constantino |
| National Institutes of Health | 1S10RR029274-01 | Meng Wu |
| Associazione Italiana per la Ricerca sul Cancro | AIRC, IG13398 | Pietro Pichierri |
| Nando and Elsa Peretti Foundation | 2012-113 | Pietro Pichierri |
| National Institutes of Health | R01-GM097373 | M Ashley Spies |
| University of Iowa | College of Medicine pilot HTS grant | Michael Ashley Spies Maria Spies |
| American Cancer Society | RSG-09-182-01-DMC | Maria Spies |

The funders had no role in study design, data collection and interpretation, or the decision to submit the work for publication.

### Author contributions

SRH, LFdSC, LY, MW, MAS, MS, Conception and design, Acquisition of data, Analysis and interpretation of data, Drafting or revising the article; EM, Conception and design, Acquisition of data, Analysis and interpretation of data; FEB, Acquisition of data, Analysis and interpretation of data, Drafting or revising the article; AD, BGK, Acquisition of data, Analysis and interpretation of data; PP, Conception and design, Analysis and interpretation of data, Drafting or revising the article

### Author ORCIDs

Maria Spies, http://orcid.org/0000-0002-7375-8037

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
