## [Decision Letter]

[Editors’ note: this article was originally rejected after discussions between the reviewers, but the authors were invited to resubmit after an appeal against the decision.]

Thank you for submitting your work entitled "Small-molecule inhibitors identify the RAD52-ssDNA interaction as critical for survival of BRCA2 deficient cells" for consideration by *eLife*. Your article has been reviewed by three peer reviewers, and the evaluation has been overseen by a Reviewing Editor and Charles Sawyers as the Senior Editor. Our decision has been reached after consultation between the reviewers. Based on these discussions and the individual reviews below, we regret to inform you that your work will not be considered further for publication in *eLife*. One of the reviewers involved in the assessment of your submission has agreed to reveal his identity: William Holloman.

As you can see below, the reviewers appreciate the work and its potential importance but raised three important issues that preclude further consideration of the manuscript for *eLife*, at least in the present form. I can see that the issues may be addressed but the required experiments seem much beyond what can be done in two months, which is a requirement for considering a revision in *eLife*. However, should you be able to address all of the major concerns, we would be willing to consider a revised manuscript with no guarantee of acceptance. Of course, given the numerous changes specified by our reviewers, you may wish to submit this work elsewhere.

Major concerns:

1) The inhibition mechanism proposed is not convincing due to several issues in cell-based assays.

2) Previous related studies reduce the impact of the current study is an important point concerning the significance of the work.

3) The molecular modeling studies may be the main novelty compared to previous studies on Rad52 inhibitors by other researchers but do not add much to the manuscript in the absence of experimental validation.

*Reviewer 1:*

Hengel et al. identified novel small molecule inhibitors of RAD52, a protein required for *BRCA1, BRCA2*, and *PALB2*-deficient cell survival. This work trails at least three other studies that have used similar HTS assays to identify RAD52 inhibitors (Huang et al., NAR, 2016; Sullivan et al., PLoS ONE, 2016; Chandramouly et al., Chem Biol 2015). This manuscript needs to address how the compounds identified in this screen compare to the compounds that were previously identified as RAD52-ssDNA binding inhibitors. Is this largely a confirmation of what has been seen previously, or does the current work uncover new molecular scaffolds? In addition, this paper largely concludes that compounds '1' and '6' disrupt RAD52-ssDNA binding. However, an alternative hypothesis may be that these compounds affect the RAD52 superstructure, as suggested in Chandreamouly et al. This hypothesis can be easily tested here. Finally, the cell biology work requires multiple controls prior to publication in *eLife*.

Major Comments:

1) Addition of RPA in Figure 3 significantly abrogates the ability of compound '6' to inhibit RAD51. A further characterization of how RPA binding affects '6' may be useful and shed light on the in vivo effects of '6' (See also Figure 5). Perhaps titrating the RPA concentration in Figure 3 will have a dramatic effect on the IC50, whereas the same titration in Figure 2 will not. Careful analysis of this effect will assess the utility of '6' for future studies.

2) Compound '6' was dropped from Figure 6 and Figure 7, perhaps because of a high IC50 in Figure 5. A comment on this in the text would improve readability and explain the disappearance of '6' from the rest of the manuscript.

The Discussion should also mention the RPA binding to '6'. Perhaps this is a reason for its high IC50 in cells.

3) Figure 6 shows that compound '1' induces cell death when cells are lacking MUS81 or BRCA2; however, the controls are a bit misleading. Knockdown of MUS81 in Figure 6 is clearly decreasing the amount of BRCA2 in cells, suggesting that BRCA2 alone may be causing the effects seen in this figure. It may be appropriate to either overexpress BRCA2 or otherwise use other siRNAs in order to retain BRCA2 to claim that MUS81 is needed for survival in the presence of this compound. Furthermore, it would be interesting to see if double knockdown of BRCA2 and MUS81 had an additional effect on survival. Also, in Figure 6, the t-tests have been performed between control knockdown cells and MUS81/BRCA2 knockdown cells with and without inhibitor. The more appropriate comparison would be between knockdown cells without and with inhibitor. Finally, Figure 6 is not very convincing. Perhaps a higher magnification or contrasted image would be better.

4) Figure 7 does not include a knockdown of RAD52 alone, perhaps because it also affects BRCA2 levels. Figure 7 suffers from largely the same problems as Figure 6. An effort to increase the quality of these cellular studies would greatly improve this manuscript and tie in nicely with the excellent FRET-based work. Figure 7 shows a very nice example of the cellular assay, unlike Figure 6.

5) The authors could further test the method of action of their drugs by attempting to observe RAD52 foci in both stressed and unstressed cells. Presumably the drugs should reduce DNA damage-induced RAD52 foci.

*Reviewer #2:*

Hengel et al. present the results of a drug screen for compounds that inhibit the ability of RAD51 to bind ssDNA. A number of hit compounds are found and of these two characterized biochemically and in cell based assays. Evidence for direct binding of the two compounds was obtained by a method that I am unfamiliar with (WaterLOGSY) and computational analysis is used to identify potential binding sites. One of the lead compounds '1' is found to have inhibitory activity on DNA cleavage activity that depends on MUS81 and RAD52 in tissue culture cells. Analysis of the effects of the compound on cell viability indicates that the compound has modest toxicity in both the presence and absence of BRCA2 and MUS81. The compound also modestly enhances the toxicity of HU.

Overall the experiments are appropriate and of good quality (I admit I am not an expert on NMR or computational modeling of binding interactions). However, there are two particularly disappointing aspects of the results presented, as described below. In addition, I do not feel that the impact of this work justifies publication in *eLife*.

Problems with experiments or interpretations:

First, although the molecular modeling studies seem promising, they are of little impact without experimental verification using mutant proteins altered at residues modeled to be critical for binding. Such mutants should be generated and tested. If an attempt to do this has been unsuccessful, that obviously raises serious concern regarding the validity of the computational work.

Secondly, although the COMET assays do support the notion that compound '1' has inhibitory activity on RAD52 in cells, the cell viability assays do not connect the toxicity of the drug with replication stress or homologous recombination. Combined effects are essentially additive for BRCA2 depletion and for HU treatment. This is the expected result if the mechanism of toxicity of '1' and '6' is completely unrelated to any impact on replication stress or HR.

One obvious missing control is a RAD52 knock-down in combination with '1" treatment. If the effect of '1' on viability is mediated by inhibition of RAD52, the combination should be no more toxic than the effect of siRNA RAD52 or "1" treatment alone (assuming siRNA treatment is effective). This control is provided in the COMET assay, but not the cell viability assays. Given that the authors discuss the importance of this control for the COMET assay data, it is not clear why it was omitted from the viability assays.

Setting aside the shortcomings described above, the work does not provide significant biological or biochemical insight with respect to the mechanisms of replication stress tolerance or homologous recombination. Furthermore, if PARP inhibitors set a standard for what would constitute an influential approach to targeting BRCA2-defective tumor cells, the new compounds to not meet and certainly do not exceed that standard. Thus, while the work is nicely done and will certainly be of interest to some members of the replication and homologous recombination communities, I do not think *eLife* is the right place for it.

*Reviewer #3:*

Here the investigators have identified small molecule inhibitors of RAD52 with the goal of utilizing these in therapeutic treatment of cancers with defective homologous recombination genes. Using a FRET based assay for RAD52 DNA binding the investigators screened a small library of compounds and obtained several candidates. Two were characterized for effects on RAD52 promoted annealing of complementary strands, interaction with RPA, self-interaction, computational docking to the putative DNA binding groove in the RAD52 undecamer structure, and several cell based assays. The work is well-executed, the quality of the data is high, and the work is well written. The findings will be of interest to investigators in the field and there is potential for translational application.

I have a few suggestions for the investigators to consider.

1) Richard Pomerantz and colleagues recently reported identification of small molecule inhibitors of RAD52 using a different assay. That work should be cited.(Chem Biol 22, 1491 2015).

2) It would be confirmative to show the IC50s of the compounds on DNA binding by gel shift assay.

3) Figure 2—figure supplement 2. There should be a control for effects of compound on spontaneous annealing. Spontaneous annealing can be fast. One might think that the inhibitors would reduce the rate to that level. Is it really zero as shown in the figure? Or has that rate been subtracted out?

4) Figure 5. As the determination is aimed at examining RAD52/MUS81 mediated DSB formation, siMUS81 comet assay controls should be performed. It would also be interesting to show that the effect of the compounds is specific for RAD52 by doing a RAD51 siRNA.

5) Figure 6. Please explain briefly in the methods the basis of the LIVE/DEAD assay. Traditionally most graphs detailing cell killing are presented as% survival.% death seems backwards to me.

6) Figure 7. The legend title is incorrect (same as Figure 6). Controls are needed for addition of compound plus siRAD52 and also siBRCA2.

[Editors’ note: what now follows is the decision letter after the authors submitted for further consideration.]

Thank you for resubmitting your work entitled "Small-molecule inhibitors identify the RAD52-ssDNA interaction as critical for survival of BRCA2 deficient cells" for further consideration at *eLife*. Your revised article has been favorably evaluated by Charles Sawyers (Senior editor), a Reviewing editor, and three reviewers.

The manuscript has been improved but there are some remaining issues that need to be addressed before acceptance. Reviewer 2 has concerns about the paper's suitability for *eLife*, and while I concur with the other reviewers that your manuscript already has substantial amount of novel data and analysis that justify its publication in *eLife*, I request that you add a short paragraph to discuss limitations of your present study and what can be done in the future to further validate the approach.

*Reviewer #1:*

This revised version had addressed my main concerns. While I agree with reviewer 2 that mutating multiple RAD52 residues would be a good way to validate the drug-interacting surface, this may also destabilize the protein or its multimers. I can see why the authors want to avoid going down that rabbit hole. Using the computational pipeline to discover another drug indicates that it has some predictive potential. This makes it a valuable tool in its own right. I am also satisfied by the additional cell biology results and a more thorough comparison with inhibitors that have been published by other groups. In sum, I think this paper is ready for publication in *eLife*.

*Reviewer #2:*

The revised manuscript addressed one of my concerns (a control showing the compound '1' did not increase lethality in a RAD52 knock-down context). However, two major issues prevent me from changing my mind with respect to the suitability of this paper for *eLife*.

First, there is still no verification of the binding site by mutational analysis. The authors claim this is not possible because of the nature of the binding site. I agree that single point mutants may not be sufficient to detect an effect on binding, but this is not an impediment as it is just as easy to make multiple mutations, that lack several of the hypothesized contacts, as it is to make point mutants. A handful of double or triple mutants could have been designed (with the help of the in silico tools) and tested for effects on binding affinity.

Second, as mentioned previously, the in vivo work reveals only modest additive effects and not the synergy required to make '1' a particularly attractive candidate to develop for treatment of BRCA2 patients.

The new material added on in silico screening is interesting and has potential to be very important. However, again, there is no mutant validation. Furthermore, the compound that came out of the screen binds to RPA. Hence, it seems possible that NP-00425 is a relatively promiscuous binder and/or that the binding to RPA will result in problematic off pathway activity in vivo. There was no in vivo work done on the new compound to address this possibility. Furthermore, the binding to RPA raises the possibility that the compound binds a surface of RAD52 other than that predicted by the constraints of the in silico approach.

I appreciate that the in silico method described COULD have high impact, but one will need to see if the success reported here is validated by demonstration that the hits are actually binding as expected. Even with mutational or other validation, one would want evidence that the approach is generally applicable to justify publication of the paper in *eLife* on the basis of this new method.

I still feel it is unlikely that this paper is going to be "highly influential" relative to related studies that have been published in recent years. It is a very nice paper, but not up to *eLife*'s standards as I understand them.

*Reviewer #3:*

This is a revised version of a manuscript that reported identification of inhibitors of Rad52 from a HTS. The issues that I had raised have been adequately addressed and from my perspective the manuscript has been tightened up and much improved. In addition, this revised version has an added new section on validation using in silico screening that led to discovery of novel lead compounds with different chemical spaces. While this approach is well beyond my area of expertise, it nonetheless seems clear that this is an authoritative analysis that adds a new level of innovation to the manuscript. I enjoyed reading the manuscript and find it to be a contribution that will be appreciated by investigators in the field.

---

## [Author Response]

[Editors’ note: the author responses to the first round of peer review follow.]

As you can see below, the reviewers appreciate the work and its potential importance but raised three important issues that preclude further consideration of the manuscript for eLife, at least in the present form. I can see that the issues may be addressed but the required experiments seem much beyond what can be done in two months, which is a requirement for considering a revision in eLife. However, should you be able to address all of the major concerns, we would be willing to consider a revised manuscript with no guarantee of acceptance. Of course, given the numerous changes specified by our reviewers, you may wish to submit this work elsewhere.

We carefully considered the critiques and, I believe, we can fully address the identified concerns. The reviewers raised a number of excellent points, addressing these points allowed us to significantly strengthen the manuscript. I hope you will find the changes we implemented and our responses compelling.

*Major concerns:*

*1) The inhibition mechanism proposed is not convincing due to several issues in cell-based assays.*

We have resolved the issues raised by the reviewers; see point by point responses below. Briefly, the revised manuscript now includes:

1) Better quality Western blot images, which unambiguously show that depletion of one of the investigated proteins in no case affects the level of the others;

2) Additional internal controls to exclude misinterpretation of data in the Comet and viability assays;

3) Enhanced immunofluorescence images in the live/dead viability assays, to make it easier to evaluate the key phenotypes;

4) Finally, Reviewer 1 inquired about the effect of the RAD52 inhibitor on the damage-induced and spontaneous RAD52 foci. In the point by point response below we show the results of the RAD52 immunofluorescence study that indicates that compound ‘1’ interferes with the recruitment of RAD52 at nuclear foci upon DNA damage. This result is provided for evaluation by reviewers but not included in the manuscript as it will dilute the main message of the work.

*2) Previous related studies reduce the impact of the current study is an important point concerning the significance of the work.*

We should note that the papers by Huang et al. (NAR, 2016) and Sullivan et al. (PLoS ONE, 2016) have been published while our manuscript was under consideration in *eLife*. While these papers do report the RAD52 inhibitors, they offer a limited insight into the mechanism of the identified small molecules. The third manuscript by Chandramouly et al. (published in late fall 2015) identified a class of inhibitors that acts differently from the molecules we identified, as it interferes with the RAD52 oligomerization and the supramolecular assembly. We have included an extensive discussion on the complementarity of the two studies.

The following statement has been added to the Discussion:

Recently, Chandramouly and colleagues (Chandramouly et al., 2015) identified a small-molecule RAD52 inhibitor, 6-hydroxy-DL-dopa, that acts differently from the molecules reported here. This inhibitor interferes with the RAD52 oligomerization and the supramolecular assembly by an unresolved mechanism. It may act by binding at the RAD52 monomer-monomer interface, or at a different site on the protein and acts allosterically. The existence of the distinct classes of RAD52 inhibitors, exemplified by ‘1’ and 6-hydroxy-DL-dopa, suggests that disrupting the RAD52-ssDNA interaction or the integrity of the RAD52 oligomeric ring bears negative consequences for the RAD52 cellular functions. Considering that the efficient homology search and the DNA strand annealing requires the two complementary DNA strands (or the complementary ssDNA-RPA complexes) to be wrapped around the two different RAD52 oligomeric rings (Grimme et al., 2010, Rothenberg et al., 2008), this is not surprising, and offers an exciting opportunity for development of more specific and potent agents for targeting recombination-deficient tumors.

While this manuscript was under review, two studies reporting small-molecule inhibitors of RAD52 were published. In the first study, Huang et al. (Huang et al., 2016) carried out an HTS campaign to identify 17 compounds that inhibit RAD52-mediated annealing in vitro with IC50 values ranging between 1.7 and 17 µM, which physically bind RAD52, and selectively, albeit at high concentrations, inhibit the single-strand annealing pathway of DSB repair over homologous recombination. In another study, Sullivan et al. (Sullivan et al., 2016) reported an in silico docking screen of a library of drug-like compounds. Among 36 predicted small-molecules, 9 compounds inhibited RAD52-ssDNA interaction in vitro, and 1 in cells. As with all previous publications, the authors screened different libraries, resulting in compounds that represent a different chemical space from those that emerged from our campaign. In addition to the high lipophilicity and relatively poor ligand efficiency of inhibitors discovered in previous campaigns, our work is deeply linked with solid structure-activity relationships based on the RAD52-ligand complexes (i.e. the current work moves from phenomenology to a definitive workflow to exploit the ssDNA binding pocket). Additionally, the use of large scale natural product screening using an in silico approach is highly innovative, and has yielded a successfully hit, which provides an enormous opportunity utilize this pharmacophore in future campaigns.

*3) The molecular modeling studies may be the main novelty compared to previous studies on Rad52 inhibitors by other researchers but do not add much to the manuscript in the absence of experimental validation.*

We very much appreciate that the reviewers see the novelty and the potential insight offered by our computational studies. We agree that the experimental validation is needed to confirm the compound placement. However, the reason we did not carry out the analysis of a point mutant that would interfere with the compound binding is due to the nature of the compound-RAD52 complexes. The majority of the contacts are achieved through a shallow continuum of van der Waals interaction, and many are mediated by water molecules (i.e., the complexation is not analogous to classic enzyme and receptor pockets). Indeed, the very nature of the ssDNA binding groove presents unique challenges to direct attempts to parse the binding free energy contributions from the amino acid moieties. Our binding energy calculations suggest that none of the possible single mutants will have a significant effect on the compound binding, at least without destabilizing the protein. We have, however, another and more powerful validation of our compound placement: We applied our compound placement protocol (docking of the small molecules to the “hotspots” along the ssDNA binding groove of RAD52 followed by refinement of receptor-ligand complexes with an all-atom simulated annealing energy minimization and ligand free energy re-scoring approach, using a knowledge-based force field in explicit solvent) to virtually screen a larger library of natural products. The campaign yielded 9 promising compounds, all very distinct in structure from our original hits. Extraordinarily, the natural product compound NP-00425 that emerged from the in silico screen of ~ 5K compounds and was selected for in vitro testing, yielded a promising hit that binds to RAD52 and competes with ssDNA binding, as confirmed by both our biophysical assays as well as NMR WaterLOGSY experiments. The strength of our method is that the docking, scoring and selection protocol was designed to enrich the true positive hits (the docked validated hits) and decrease the false negatives and false positives. Importantly, this manuscript fully describes an integrated and highly complementary HTS/in silico screening approach that worked wonderfully in producing structure based lead compounds. In short, the initial HTS screen provided the critical de novo structure activity relationship, while the in silico method clearly elucidated the key metrics that describe complexation in the ssDNA binding groove. These studies will provide a strong foundation for the discovery of novel antineoplastic therapeutics. We believe that the success of this virtual screening campaign, not only validates the original compound placement, but goes a step further, and demonstrates the enormous success of such an approach in producing bone fide hits in this exceedingly difficult target class.

The description of these studies was added as the following new sections:

Results:

In Silico Screening and Discovery of NP-00425: translating structure-activity relationships from HTS into novel inhibitors

Natural product NP-00425 physically interacts with RAD52 and RPA proteins, inhibits the RAD52 binding to ssDNA and the ssDNA-RPA complex, but does not affect RAD52-dsDNA interaction or the ssDNA binding by RPA.

The relevant sections in the Introduction and Discussion were also updated.

*Reviewer #1:*

*Hengel et al. identified novel small molecule inhibitors of RAD52, a protein required for BRCA1, BRCA2, and PALB2-deficient cell survival. This work trails at least three other studies that have used similar HTS assays to identify RAD52 inhibitors (Huang et al., NAR, 2016; Sullivan et al., PLoS ONE, 2016; Chandramouly et al., Chem Biol 2015). This manuscript needs to address how the compounds identified in this screen compare to the compounds that were previously identified as RAD52-ssDNA binding inhibitors. Is this largely a confirmation of what has been seen previously, or does the current work uncover new molecular scaffolds?*

As mentioned above, the studies by Huang et al. and Sullivan et al. were published while this manuscript was under consideration in *eLife*. In the revised version we added a discussion of the compounds identified in both studies in terms of both scaffolds and potency (see comments above).

Specifically, the high lipophilicity and relatively poor ligand efficiency of inhibitors discovered in previous campaigns suggests that novel scaffolds will be essential to take the RAD52 drug discovery to the next level. Our work is deeply linked with solid structure-activity relationships based on the RAD52-ligand complexes (i.e. the current work moves from phenomenology to a definitive workflow to exploit the ssDNA binding pocket). Additionally, the use of large scale natural product screening using an in silico approach rooted in the structure-activity relationships obtained in the analysis of the initial HTS hits is highly innovative, and has yielded a successfully hit with a completely novel scaffold. This methodology provides an enormous opportunity of utilizing this pharmacophore in future campaigns.

*In addition, this paper largely concludes that compounds '1' and '6' disrupt RAD52-ssDNA binding. However, an alternative hypothesis may be that these compounds affect the RAD52 superstructure, as suggested in Chandreamouly et al. This hypothesis can be easily tested here. Finally, the cell biology work requires multiple controls prior to publication in eLife.*

While all experimental evidence in our study points towards the inhibition of the RAD52-ssDNA interaction, we have considered the possibility that '1' and '6' (or any of our HTS hits) may also interfere with the oligomeric or supramolecular structure of RAD52. The dynamic light scattering analysis (Figure 2—figure supplement 3), however, rules out a possibility that our inhibitors act by disrupting the RAD52 rings, as no change in the RAD52 oligomeric state is detected in the equimolar presence of these compounds.

To the original text “Dynamic light scattering experiments conducted in the presence of equimolar concentrations of each compound and RAD52 showed that the presence of these compounds neither breaks up the oligomeric ring of RAD52 nor causes protein aggregation (Figure 2—figure supplement 3).” We added the following statement: “Notably, this means that our compounds act differently from the RAD52 inhibitor 6-hydroxy-DL-dopa (Chandramouly et al., 2015), which disrupts supramolecular assembly of the RAD52 protein.”

*Major Comments:*

*1) Addition of RPA in Figure 3 significantly abrogates the ability of compound '6' to inhibit RAD51. A further characterization of how RPA binding affects '6' may be useful and shed light on the* in vivo *effects of '6' (See also Figure 5). Perhaps titrating the RPA concentration in Figure 3 will have a dramatic effect on the IC50, whereas the same titration in Figure 2 will not. Careful analysis of this effect will assess the utility of '6' for future studies.*

The positive WaterLOGSY peaks (Figure 3) indicate that ‘6’ interacts with RPA, though it does not disrupt the RPA-ssDNA interaction (Figure 3); the interaction between ‘6’ and RAD52 is promiscuous as it disrupts the RAD52-dsDNA interaction (Figure 3). This promiscuity in the interactions with multiple components of the reaction may be responsible for the reduced IC50 value for the inhibition of the ssDNA-RPA binding by RAD52. The nearly 5-fold change in the potency is, however, negated by the fact that the effect of ‘6’ on the strand annealing reaction is within experimental error for the protein-free and RPA-coated ssDNA (Figure 3).

*2) Compound '6' was dropped from Figure 6 and Figure 7, perhaps because of a high IC50 in Figure 5. A comment on this in the text would improve readability and explain the disappearance of '6' from the rest of the manuscript.*

*The Discussion should also mention the RPA binding to '6'. Perhaps this is a reason for its high IC50 in cells.*

We agree with the reviewer’s conclusion and suggestion. The following statement in the Results section “Higher concentration of ‘6’ required to inhibit MUS81/EME1/RAD52-mediated DSBs (Figure 5) is likely due to the particular chemical nature of this compound, which is a promiscuous binder; not only does it interact with RPA and binds within the dsDNA binding site of RAD52, but has been identified as an inhibitor in 192 different HTS assays (Pubchem).” Has been expanded with the following clarification: “Due to its lower capacity to inhibit the MUS81/EME1/RAD52-mediated DSBs, and its expected off-target effects, we have eliminated ‘6’ from further analysis and focused all our subsequent cellular studies on ‘1’, appeared more specific in our biochemical studies and had no Pubchem hits.”

3) Figure 6 shows that compound '1' induces cell death when cells are lacking MUS81 or BRCA2; however, the controls are a bit misleading. Knockdown of MUS81 in Figure 6 is clearly decreasing the amount of BRCA2 in cells, suggesting that BRCA2 alone may be causing the effects seen in this figure. It may be appropriate to either overexpress BRCA2 or otherwise use other siRNAs in order to retain BRCA2 to claim that MUS81 is needed for survival in the presence of this compound. Furthermore, it would be interesting to see if double knockdown of BRCA2 and MUS81 had an additional effect on survival. Also, in Figure 6, the t-tests have been performed between control knockdown cells and MUS81/BRCA2 knockdown cells with and without inhibitor. The more appropriate comparison would be between knockdown cells without and with inhibitor. Finally, Figure 6 is not very convincing. Perhaps a higher magnification or contrasted image would be better.

It should be noted that decrease in the amount of BRCA2 in MUS81-depleted cells was only perceived because the amount of total protein in the siMUS81-depleted cells was lower than in the other lanes. Indeed, if normalized against LMNB1, the amount of BRCA2 was similar in the non-BRCA2 RNAi experimental points. To improve quality of the data, however, we repeated the viability assay and presented a better-quality Western blot (all lanes are now balanced) to show efficiency and specificity of the RNAi reagents used in the assay. This new Western blot clearly shows that MUS81 knock-down does not affect BRCA2 levels. Please, note that as the inhibitor does not affect the level of the investigated DNA repair proteins, for sake of clarity, we do not show in the revised Western blot image samples treated with RNAi and the RAD52 inhibitor.

In the revised manuscript, we also included the analysis of cell death in double MUS81+BRCA2 knock-down cells. Interestingly, concomitant depletion of BRCA2 and MUS81 strongly increases cell death even in the absence of treatment. The lethality in the untreated cells co-depleted of MUS81 and BRCA2 is elevated compared to BRCA2-depleted cells with depleted or inhibited RAD52. This is not unexpected. Although we have previously shown that MUS81 and RAD52 can cooperate in supporting the replication fork recovery in the checkpoint-inhibited cells (Murfuni et al., 2013), MUS81 has multiple RAD52-independent functions, such as resolution of intermediates in M-phase. Loss of such MUS81 mitotic function may impact the viability of BRCA2-deficient cells. These new data may imply that BRCA2-deficent cells have to engage the MUS81-RAD52 route to deal with the demised replication forks, and, on the other hand, point to MUS81 as another potential target for small-molecule inhibitors to selectively kill BRCA2-deficent tumor cells.

We now performed statistical analysis comparing each pair of experimental points, in the presence and absence of the inhibitor.

Live/dead assay is analyzed blindly after acquisition of the images at a low magnification in order to collect a large population of the cells. So, it was not feasible to include high magnification images. However, we increased the contrast and brightness of all images to make easier identification of dead cells, which are red-stained, in the live population, which is green-stained. We also provide a more extended explanation in the methods that the percentage of cell death reported in the graph derives from calculation of the number of dead cells in the population but it is also normalized against the number of fields required to score the number of cells required in the assay. This normalization is performed in order to include in the number of dead cells also those that are floating in the well and are inevitably lost during the staining protocol. So, a very low number of green, live, cells, may end up in elevated percentage of cell death even in the presence of few red, dead, cells.

4) Figure 7 does not include a knockdown of RAD52 alone, perhaps because it also affects BRCA2 levels. Figure 7 suffers from largely the same problems as Figure 6. An effort to increase the quality of these cellular studies would greatly improve this manuscript and tie in nicely with the excellent FRET-based work. Figure 7 shows a very nice example of the cellular assay, unlike Figure 6.

As already stated for the revised Figure 6, we now included a new Western blot image and performed additional experiments, including those using RAD52 siRNAs, to show that depletion of RAD52 does not affect BRCA2 levels and that, as also reported in the revised Figure 6, inhibition of RAD52 in cells depleted of RAD52 does not increase cell death further as compared with each single condition.

*5) The authors could further test the method of action of their drugs by attempting to observe RAD52 foci in both stressed and unstressed cells. Presumably the drugs should reduce DNA damage-induced RAD52 foci.*

As for the reviewer’s suggestion, we analyzed the formation of RAD52 foci after DNA damage in the presence and in the absence of ‘1’. The figure summarizing the results is included below. We did not, however, include these data in the manuscript because this analysis goes beyond the scope of the current paper. The analysis of the RAD52 foci formation along with the effect of RAD52 inhibition on the HR, SSA, and other DNA repair pathways will comprise another detailed study, which will follow up this current manuscript and will require extensive work to verify the mechanism of the inhibitor action in the cells (use of cells expressing RAD52 that cannot interact with RPA-ssDNA, for instance). These analyses will deviate too much from the scope and main message of the current work.

As shown in Figure 10 in response to DSBs, RAD52 accumulation in the nuclei is strongly reduced by treatment with the inhibitor. Additionally, the reduction in the number of RAD52-positive nuclei is observed after treatment. Thus, inhibition of RAD52-ssDNA association is sufficient to negatively affect the RAD52 foci-formation upon DNA damage. Unfortunately, we cannot speak about the effect in untreated cells as few RAD52-positive nuclei are detectable in the absence of DNA damage.

Author response image 1.The RAD52 inhibitor interferes with the ability of RAD52 to localize in nuclear foci after DSBs.**DOI:**
http://dx.doi.org/10.7554/eLife.14740.018

hTert-immortalized primary human fibroblasts were transfected where indicated with RAD52 siRNAs, and 48h post-transfection treated with etoposide for 3h. After treatment cells were subjected to indirect immunofluorescence using anti-RAD52 primary antibody. Immunofluorescence was performed by in situ extraction to retain only chromatin-bound proteins. The number of RAD52-foci-positive nuclei was recorded, and the intensity of fluorescence evaluated using imageJ. Fluorescence intensity data are presented as the mean of the total number of positive nuclei expressed as arbitrary units after background normalization. Note that the inhibitor also qualitatively affects the RAD52 immunostaining with little persistence of well-defined foci in the nucleus.

*Reviewer #2:*

*Hengel et al. present the results of a drug screen for compounds that inhibit the ability of RAD51 to bind ssDNA. A number of hit compounds are found and of these two characterized biochemically and in cell based assays. Evidence for direct binding of the two compounds was obtained by a method that I am unfamiliar with (WaterLOGSY) and computational analysis is used to identify potential binding sites. One of the lead compounds '1' is found to have inhibitory activity on DNA cleavage activity that depends on MUS81 and RAD52 in tissue culture cells. Analysis of the effects of the compound on cell viability indicates that the compound has modest toxicity in both the presence and absence of BRCA2 and MUS81. The compound also modestly enhances the toxicity of HU.*

*Overall the experiments are appropriate and of good quality (I admit I am not an expert on NMR or computational modeling of binding interactions). However, there are two particularly disappointing aspects of the results presented, as described below. In addition, I do not feel that the impact of this work justifies publication in eLife.*

*Problems with experiments or interpretations:*

*First, although the molecular modeling studies seem promising, they are of little impact without experimental verification using mutant proteins altered at residues modeled to be critical for binding. Such mutants should be generated and tested. If an attempt to do this has been unsuccessful, that obviously raises serious concern regarding the validity of the computational work.*

We agree that the experimental validation is needed to confirm the compound placement. However, the reason we did not carry out the analysis of the point mutants that would interfere with the compound binding is due to the nature of the compound-RAD52 complexes. The majority of the contacts are achieved through a shallow continuum of van der Waals interaction, and many are mediated by water molecules (i.e., the complexation is not analogous to classic enzyme and receptor pockets). Indeed, the very nature of the ssDNA binding groove presents unique challenges to direct attempts to parse the binding free energy contributions from the amino acid moieties. Our binding energy calculations suggest that none of the possible single mutants will have a significant effect on the compound binding, at least without destabilizing the protein. We have, however, another and more powerful validation of our compound placement:

This revised manuscript describes the implementation of a highly robust method for designing a hybrid in silico/in vitro screening, which successfully employed a library of natural products compounds. This campaign resulted in the discovery of macrocycle inhibitor of RAD52 (NP-00425). Extraordinarily, the natural product compound that emerged from the in silico screen (of ~ 5K compounds), and was selected for in vitro testing, yielded a promising hit that binds to RAD52 and competes with ssDNA binding, as confirmed by both our biophysical assays as well as NMR WaterLOGSY experiments. The strength of our method is that the docking, scoring and selection protocol was designed to enrich the true positive hits (the docked R-compounds) and decrease the false negatives and false positives. Importantly, this manuscript fully describes an integrated and highly complementary HTS/in silico screening approach that worked wonderfully in producing structure based lead compounds. In short, the initial HTS screen provided the critical de novo structure activity relationship, while the in silico method clearly elucidated the key metrics that describe complexation in the ssDNA binding groove. These studies will provide a strong foundation for the discovery of novel antineoplastic therapeutics. Additionally, we feel that the discovery of NP-00425 and its ability to compete with ssDNA in a ssDNA binding groove of RAD52 will generate significant interest in a broad range of disciplines, and strongly enhances the impact of the current study.

*Secondly, although the COMET assays do support the notion that compound '1' has inhibitory activity on RAD52 in cells, the cell viability assays do not connect the toxicity of the drug with replication stress or homologous recombination. Combined effects are essentially additive for BRCA2 depletion and for HU treatment. This is the expected result if the mechanism of toxicity of '1' and '6' is completely unrelated to any impact on replication stress or HR.*

The effect of RAD52 inhibition is additive with either HU-induced replication stress or with BRCA2 depletion, and these additive effects are then additive themselves. This phenomenon may be somehow related to the response to replication stress. One can consider the DSBs suppression and toxicity of the RAD52 inhibitor as two connected events. Indeed, as proposed by a seminal Genes&Dev paper from the Venkitaraman’s group, loss of BRCA2 results in the fork collapse because of accumulation of pathological replication intermediates. Recently, these results were corroborated by many groups, linking BRCA2-RAD51-FANCs to fork stabilization. The pathological replication intermediates left unprotected in the BRCA2 absence may be targeted by MUS81 in collaboration with RAD52, as may happen to those accumulating in BRCA2 wild-type cells experiencing replication stress induced by checkpoint inhibition or prolonged replication arrest (Figure 6 and Figure 7, respectively).

*One obvious missing control is a RAD52 knock-down in combination with '1" treatment. If the effect of '1' on viability is mediated by inhibition of RAD52, the combination should be no more toxic than the effect of siRNA RAD52 or "1" treatment alone (assuming siRNA treatment is effective). This control is provided in the COMET assay, but not the cell viability assays. Given that the authors discuss the importance of this control for the COMET assay data, it is not clear why it was omitted from the viability assays.*

We agree. The revised Figure 6 and Figure 7 now include the analysis of cell death in the cells depleted of RAD52 in the presence or absence of ‘1’. As expected, and in agreement with the Comet assay data, the concomitant depletion of RAD52 and treatment with “1” fails to induce any additive effect.

Setting aside the shortcomings described above, the work does not provide significant biological or biochemical insight with respect to the mechanisms of replication stress tolerance or homologous recombination. Furthermore, if PARP inhibitors set a standard for what would constitute an influential approach to targeting BRCA2-defective tumor cells, the new compounds to not meet and certainly do not exceed that standard. Thus, while the work is nicely done and will certainly be of interest to some members of the replication and homologous recombination communities, I do not think eLife is the right place for it.

We believe that the addition of the new data, and in particular the discovery of a new RAD52 inhibitor based on the computational workflow, offers an important insight into the mechanisms of RAD52 inhibition. In order to validate the strength of our hypothesis about the structural nature of the ligand-RAD52 complex, we developed a validated in silico screening campaign, based on our HTS results. We describe the discovery of NP-004255, a macrocyclic compound, which we show by NMR WaterLOGSY and biophysical assays to be a completely novel and effective inhibitor of the RAD52-ssDNA interaction. The implication of these findings for the discovery of novel therapeutics that specifically inhibit the activity of RAD52 is discussed in the revised manuscript.

Significance of our structural understanding of the RAD52 inhibitors is twofold. First, it has an important implication for the discovery and design of the novel therapeutics targeting the protein-DNA interactions; second, it confirms that the RAD52-ssDNA interaction plays a role in allowing the BRCA-deficient cells to survive. The latter was not obvious, as a coherent picture of what are the roles of human (vertebrate) RAD52 in genome maintenance is lacking. The inhibitors we identified, and in particular ‘1’, which appears selective for the ssDNA binding groove of the RAD52 ring, will be a valuable tool for dissecting the RAD52-dependent pathways.

We agree that PARP inhibitors set a standard for therapeutics that can be used to treat the BRCA-deficient tumors since olaparib has been approved for treatment of some recurrent platinum-sensitive ovarian cancers. There exists, however, a need for the broader range of options for the precision treatment of the HR deficient tumors, and RAD52 is currently one of the most coveted targets. While we are not currently offering a solution ready for trials, we provide an avenue and the tools for development of the new drug-like compounds meeting the standards of efficacy like that of the PARP inhibitors, and for dissecting the RAD52-dependent cellular pathways, which may lead to the identification of the new anticancer targets.

In addition to the use of these compounds to dissect RAD52-dependent cellular pathways, there may be significant utility in therapeutics developed from these studies. This is because, the cocktail therapies are the most effective ways of mitigating the evolution of drug resistance, especially in the antineoplastic therapies, and the RAD52-dependent pathways that allow BRCA-deficient cells to survive are likely overlapping with the pathway targeted by the PARP inhibitors.

*Reviewer #3:*

*Here the investigators have identified small molecule inhibitors of RAD52 with the goal of utilizing these in therapeutic treatment of cancers with defective homologous recombination genes. Using a FRET based assay for RAD52 DNA binding the investigators screened a small library of compounds and obtained several candidates. Two were characterized for effects on RAD52 promoted annealing of complementary strands, interaction with RPA, self-interaction, computational docking to the putative DNA binding groove in the RAD52 undecamer structure, and several cell based assays. The work is well-executed, the quality of the data is high, and the work is well written. The findings will be of interest to investigators in the field and there is potential for translational application.*

*I have a few suggestions for the investigators to consider.*

*1) Richard Pomerantz and colleagues recently reported identification of small molecule inhibitors of RAD52 using a different assay. That work should be cited.(Chem Biol 22, 1491 2015).*

We have included discussion of the differences between our compounds and the small molecule identified by Chandramouly and colleagues – please see our response to the editor above along with the discussion of the compounds identified in two studies that were published while this manuscript was under review. We believe that the results of our studies are complementary and provide broader opportunities for the successful drug discovery.

*2) It would be confirmative to show the IC50s of the compounds on DNA binding by gel shift assay.*

Our previous studies (Grimme et al. NAR 2010 and Honda et al. EMBOJ 2011) have unambiguously established the equivalency of the FRET-based assay and the EMSA, with an advantage of FRET-based assay being an equilibrium binding assay. Due to the nature of the dynamic RAD52-ssDNA complex, the gel shift in the EMSA assays can be achieved only upon crosslinking the protein to DNA. We believe, therefore, that the cross-linking will undermine the validity of the competition experiment. Moreover, the presence of the secondary DNA binding site on the top of the RAD52 ring may further convolute the data interpretation in EMSA experiments.

*3) Figure 2—figure supplement 2. There should be a control for effects of compound on spontaneous annealing. Spontaneous annealing can be fast. One might think that the inhibitors would reduce the rate to that level. Is it really zero as shown in the figure? Or has that rate been subtracted out?*

This is a really good point. Under our experimental conditions and on the reaction time scale, spontaneous annealing is not observed (150 sec, low salt buffer); there is no change in the spontaneous annealing rate in the presence of inhibitors, so the inhibitor only signal has not been subtracted. The same applies to the annealing of the RPA-ssDNA complexes.

We have added the spontaneous annealing data to the Figure 2—figure supplement 2 graph.

*4) Figure 5. As the determination is aimed at examining RAD52/MUS81 mediated DSB formation, siMUS81 comet assay controls should be performed. It would also be interesting to show that the effect of the compounds is specific for RAD52 by doing a RAD51 siRNA.*

This is a valid control, which we have previously performed. In the (Murfuni et al., 2013) manuscript we demonstrated that RAD51 depletion does not affect formation of the MUS81-dependent DSBs. We therefore consider of little relevance to analyze the effect of RAD52 inhibition in RAD51-depleted cells.

*5) Figure 6. Please explain briefly in the methods the basis of the LIVE/DEAD assay. Traditionally most graphs detailing cell killing are presented as% survival.% death seems backwards to me.*

In the revised Materials and methods section we have included a more detailed description of the assay. Briefly, the assay evaluates the number of dead cells, red, in the population of live cells, visualized as green. That is the reason why results are presented as percentage of cell death.

6) Figure 7. The legend title is incorrect (same as Figure 6). Controls are needed for addition of compound plus siRAD52 and also siBRCA2.

We apologize for the oversight. We now amended the legend title and included the analysis of cell death in cells depleted of RAD52, alone and combined with the inhibitor. Evaluation of cell death in siBRCA2 cells was present in the original figure.

[Editors’ note: the author responses to the re-review follow.]

The manuscript has been improved but there are some remaining issues that need to be addressed before acceptance. Reviewer 2 has concerns about the paper's suitability for eLife, and while I concur with the other reviewers that your manuscript already has substantial amount of novel data and analysis that justify its publication in eLife, I request that you add a short paragraph to discuss limitations of your present study and what can be done in the future to further validate the approach.

As you requested, the following paragraph has been added to the Discussion section of the revised manuscript:

“It has to be noted, however, that although our work provides a solid understanding of how shape complementarity and utilization of interstitial water networks drives complexation, there are limitations to these approaches. Specifically, the current experimental and computational studies do not address the underlying thermodynamics and kinetics that control ligand competition in the ssDNA binding groove. A fascinating aspect of the competition between small molecules and macromolecules for an extended and partially solvated binding pocket, as in RAD52, is the extent to which small molecules may exploit the potential enthalpy-entropy compensation of a large and floppy native DNA ligand. Additionally, residence times of lead compounds may be a critical factor in the successful design of therapeutic lead compounds. These factors will be addressed in the future studies that will be focused on the thermodynamics and kinetics of small molecule binding to the RAD52 protein, and in the studies that will confirm the ligand placement through high resolution structures.”